# *LogitSpec*: Accelerating Retrieval-based Speculative Decoding via Next Next Token Speculation

## Abstract

Speculative decoding (SD), where a small draft model is employed to propose *draft* tokens in advance and then the target model validates them in parallel, has emerged as a promising technique for LLM inference acceleration. Many endeavors to improve SD are to eliminate the need for a draft model and generate draft tokens in a retrieval-based manner in order to further alleviate the drafting overhead and significantly reduce the difficulty in deployment and applications. However, retrieval-based SD relies on a matching paradigm to retrieve the most relevant reference as the draft tokens, where these methods often fail to find matched and accurate draft tokens. To address this challenge, we propose *LogitSpec* to effectively expand the retrieval range and find the most relevant reference as drafts. *LogitSpec* is motivated by the observation that the logit of the last token can not only predict **the next token**, but also speculate **the next next token**. Specifically, *LogitSpec* generates draft tokens in two steps: (1) utilizing the last logit to speculate the next next token; (2) retrieving relevant reference for both the next token and the next next token. *LogitSpec* is training-free and plug-and-play, which can be easily integrated into existing LLM inference frameworks. Extensive experiments on a wide range of text generation benchmarks demonstrate that *LogitSpec* can achieve up to $2.61\times$ speedup and 3.28 mean accepted tokens per decoding step.

## 1 Introduction

Large Language Models (LLMs), such as GPT-4.5 (OpenAI, 2024), DeepSeek R1 (DeepSeek-AI, 2025), Qwen 2.5 (Team, 2025), and LLaMA-3 (Team, 2024), have demonstrated remarkable capabilities across a wide range of natural language processing tasks, including question answering (Calijorne Soares & Parreiras, 2020), code generation (Jiang et al., 2024), and dialogue systems (Yi et al., 2024). While they achieve success by scaling up the model parameters, the increase in scale comes with significant inference challenges. The most straightforward challenge is *auto-regressive decoding*, where each LLM decoding step can only generate one token. This token-by-token decoding process incurs exacerbated latency with both the length of generated tokens and the model scale.

To address this challenge, speculative decoding (SD) (Leviathan et al., 2023; Chen et al., 2023) has emerged as a promising approach for lossless LLM inference acceleration. The key idea of SD is to employ a small draft model to first generate $\gamma$ draft tokens, and then the target model validates these tokens in parallel within a single forward, ensuring the output distribution to be unchanged. However, deploying an extra draft model introduces considerable overhead and difficulties, including (a) **complex implementation and deployment**, especially in integration with other techniques,(Liu et al., 2025) (b) **additional memory overhead**, especially in long-context scenarios. Moreover, when there exists no available draft model for SD, it takes **substantial training cost** to construct a compact draft model (Li et al., 2025c). (Please refer to Section 2 for more details.)

Therefore, many existing works are dedicated to developing draft-model-free SD methods (Saxena, 2023; Fu et al., 2024; He et al., 2024), where the draft tokens are generated in a retrieval-based manner. At each decoding step, they retrieve the relevant reference **according to the last few tokens** and extract draft tokens from the reference. In this way, they can explicitly eliminate the need for a draft model and reduce the drafting overhead.

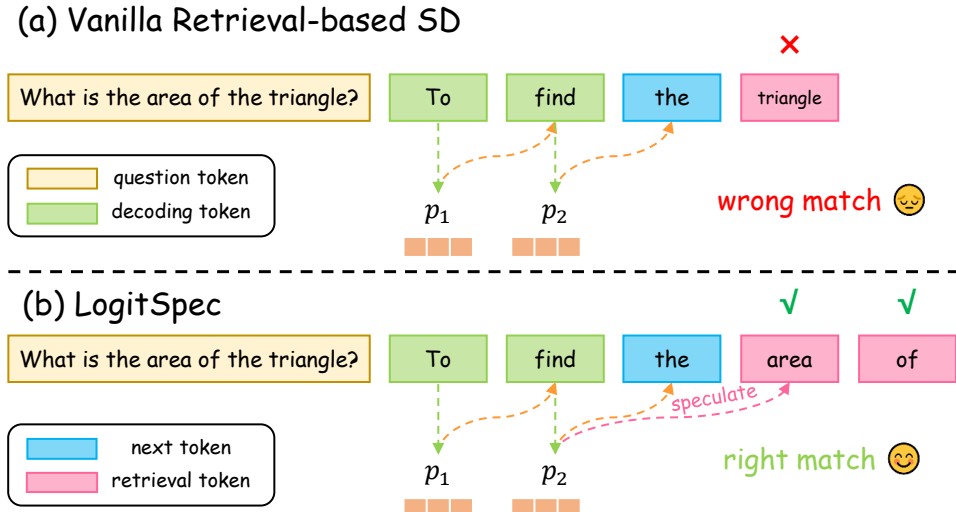

Figure 1: Illustration of vanilla retrieval-based SD method and our *LogitSpec*. (a) Retrieval-based SD retrieves the wrong token "**triangle**" according to the next token "the". (b) *LogitSpec* first speculates the next next token "area", and then retrieves the right relevant reference "**area of**" according to "the area". This simple example illustrates how *LogitSpec* utilizes the last logit to speculate the next next token and improves the retrieval accuracy.

Despite their promising efficacy and implementation simplicity, these methods rely on a matching paradigm to effectively retrieve the most relevant reference as draft tokens and often struggle to find matched and accurate tokens. As shown in Figure 1 (a), with a simple prompt "*What is the area of the triangle?*", vanilla retrieval-based SD methods fail to effectively retrieve the right token "area" according to the next token "the", as the most relevant reference is "the triangle". Moreover, it is often the case that no matched reference can be found, e.g., there are no matched tokens in more than 30% of decoding steps in PLD (Saxena, 2023).

In this paper, we propose a simple yet effective retrieval-based SD framework, namely *LogitSpec*, which leverages the logit of the last token (*last logit*) to predict the next next token and improves the accuracy of the retrieval reference. *LogitSpec* is motivated by the observation that **the last logit can speculate the next next token with a relatively high accuracy.** Based on this observation, we use the speculated next next token as guidance to retrieve reference. Specifically, *LogitSpec* generates draft tokens in two steps: (1) utilizing the last logit to speculate the next next token; (2) retrieving relevant reference for both the next token and the next next token. As shown in Figure 1 (b), with the speculated token "area", *LogitSpec* successfully retrieves the correct draft tokens "area of". This two-step process enables better retrieval accuracy—it can help filter the most relevant reference when the reference is redundant, while extending the searching space when there exists no relevant reference. To summarize, our contributions are:

(a) We empirically observe that the last logit can speculate the next next token with a relatively high accuracy. Unlike other retrieval techniques, which are sensitive to the specific task, this property is **robust** and **effective** across various tasks.

(b) Building on this observation, we propose *LogitSpec*, a **plug-and-play** retrieval-based SD framework which can improve the retrieval accuracy and achieve better speedup.

(c) We conduct extensive experiments on various text generation benchmarks to demonstrate the effectiveness of *LogitSpec*. Notably, *LogitSpec* achieves up to 2.61× speedup and 3.28 mean accepted tokens per decoding step without the need for an extra draft model.

## 2    RELATED WORK

In this section, we briefly review the two main directions of speculative decoding, draft-model-based speculative decoding and draft-model-free speculative decoding. More discussions are provided in Appendix A.

**Draft-model-based speculative decoding.**    This line of research focuses on improving the draft quality, which primarily invests heavily in post-training a specialized draft model to generate high-quality drafts. Medusa (Cai et al., 2024) pioneers this direction by adding extra MLP heads at the top of the target model, reusing the hidden states from the target model, and predicting the next few tokens in parallel. However, the generation from these decoding heads is independent, which harms the accuracy of the draft tokens. Therefore, some works (Ankner et al., 2024; Xiao et al., 2024) propose to add sequential dependencies for better performance. Glide (Du et al., 2024) takes a similar idea of Medusa and proposes to reuse the KV cache from the target model. Recently, Eagle (Li et al., 2025b; 2024; 2025c) has dominated this research line by training an auto-regressive head and scaling up the training data. **However, while these methods achieve superior speedup, all of them rely on an extra draft model, which necessitates extra parameters or extensive training.** The existence of the draft model significantly limits its application: (a) deploying a draft model requires complex and careful implementation. (b) Both the draft model and its KV cache require additional GPU memory, which may incur load imbalance. In the long context settings, even the KV cache for the draft model may exceed the capacity of 1 GPU and lead to significant resource competitions; (c) the training cost is substantial, and the draft model requires retraining when the target model is updated.

**Draft-model-free speculative decoding.**    This line of research focuses on maximizing drafting efficiency by eliminating the need for a draft model entirely, aiming for universal, zero-cost acceleration. Some existing works (Zhang et al., 2024a; Elhoushi et al., 2024; Xia et al., 2025) observe that the layer sparsity of LLMs means that not all the layers are necessary to predict the next token. They propose methods to generate draft tokens with a subset of the layers of the target model. However, the number of skipped layers is limited, which affects the overall speedup. Recently, retrieval-based SD (He et al., 2024; Saxena, 2023; Fu et al., 2024) has dominated the draft-model-free speculative decoding by retrieving the relevant reference as the draft tokens. The relevant reference can be derived from the question and generated tokens, or an external database. **However, there exist two common issues during the retrieval process**: (a) retrieval from the context often fails to find matched draft tokens; (b) retrieval from an external database often struggles to find accurate draft tokens. These two issues motivate us to develop a more appropriate retrieval-based speculative decoding framework.

## 3    BACKGROUND

**Speculative Decoding.**    Let $x$ denote an input sequence (prefix). A speculative decoding step consists of a drafting phase and a verification phase. During the drafting phase, the draft model $\mathcal{M}_q$ is employed to give $\gamma$ draft tokens $x_1, x_2, ..., x_\gamma$ by running $\gamma$ times the model forward and sampling. Here, we denote the output logit $\mathcal{M}_q(\boldsymbol{x} + [x_1, ..., x_{i-1}])$ as $q_{i-1}$, then each draft token is given by $x_i \sim q_{i-1}, i = 1, ..., \gamma$. During the verification phase, the prefix $x$ together with $\gamma$ draft tokens are sent to $\mathcal{M}_p$ for verification. The target model $\mathcal{M}_p$ inputs $\boldsymbol{x} + [x_1, ..., x_\gamma]$ and outputs the logits $p_0, p_1, ..., p_\gamma$. Then SD sequentially verifies $x_i$ via speculative sampling (Leviathan et al., 2023; Chen et al., 2023), where the acceptance rate is given by:

$$
\alpha_i = \begin{cases} 1 & p_{i-1}[x_i] \geq q_{i-1}[x_i], \\ \dfrac{p_{i-1}[x_i]}{q_{i-1}[x_i]} & p_{i-1}[x_i] < q_{i-1}[x_i], \end{cases} \tag{1}
$$

If SD rejects $x_i$, it will resample a token from $norm(\max(0, p_{i-1} - q_{i-1}))$, otherwise, SD accepts all the draft tokens and samples an additional token from $p_\gamma$. In this way, each SD step generates at least 1 token and at most $\gamma + 1$, leading to theoretical lossless quality and efficiency acceleration.

**Retrieval-based Speculative Decoding.**    The retrieval-based SD methods generate draft tokens in a retrieval-based manner. As retrieval-based SD methods do not require an additional draft model,

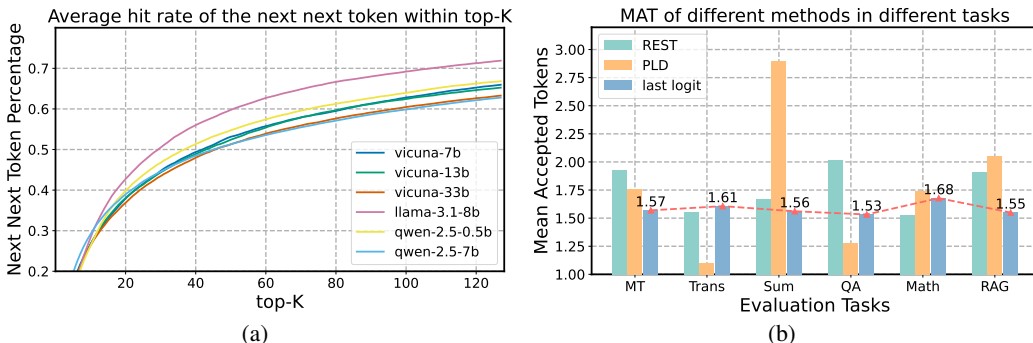

(a)

(b)

Figure 2: Motivated observations. (a) The last logit can speculate the next next token with high accuracy. In over 50% decoding steps, the next next token can be found in the top-60 entries within the last logit. This prediction ability exists consistently across different model sizes and architectures. (b) Compared with other retrieval-based methods, the prediction of the last logit demonstrates robustness to downstream tasks. Both of these observations motivate us to utilize the last logit to guide the retrieval process. **Note that the prediction of the last logit is only part of our *LogitSpec*.**

we omit the notation for $\mathcal{M}_q$ and abbreviate $\mathcal{M}_p$ as $\mathcal{M}$. A retrieval model $\mathcal{R}$ is employed to store $n$-grams (referred as **reference**) from corpus

$$\mathcal{R} = \{(x_i^1, x_i^2, ..., x_i^n)\}_{i=1}^N. \tag{2}$$

Here, $N$ denotes the total number of $n$-grams in $\mathcal{R}$. Existing retrieval-based SD methods mainly differ in constructing $\mathcal{R}$. At each SD decoding step, given a query $m$-gram tokens $x_q^1, x_q^2, ..., x_q^m$, the retrieval model first traverses the reference, finds matched $n$-grams and returns the subsequent tokens of the matched tokens:

$$\text{MATCH}(\mathcal{R}, (x_q^1, x_q^2, ..., x_q^m)) = \{(x_i^{m+1}, ..., x_i^n)\}_{i \in \mathcal{S}}, \tag{3}$$

where $\mathcal{S} = \{i | x_i^t = x_q^t, \forall t = 1, ..., m\}$. Intuitively, a larger $m$ leads to more precise matches but lower match probabilities.

## 4 METHOD

In this section, we introduce our *LogitSpec*, a novel retrieval-based SD framework that utilizes *last logit* as a guidance for retrieval and effectively improves the retrieval performance. We first conduct a motivated experiment in Section 4.1, and then introduce the whole framework of *LogitSpec* in Section 4.2 and 4.3. An overview of *LogitSpec* is shown in Figure 3.

### 4.1 MOTIVATED OBSERVATION

Drawing insights from multi-token prediction (Gloeckle et al., 2024) that fine-tuned LLMs can predict multiple tokens in a single forward, we are interested in the ability of LLMs to predict the next next token **without fine-tuning**. We conduct a simple experiment to investigate the rank of the next next token in the last logit, which is used to predict the next token. Specifically, we use a small fraction of Spec-Bench (Xia et al., 2024) that contains 13 sub-tasks (detailed in Appendix B) and 6 different LLMs as backbones. Results in Figure 2(a) demonstrate that **the last logit has a strong potential to predict the next next token**. In over 50% decoding steps, the next next token can be found within the top-60 entries of the last logit. For Llama 3.1 8B (Team, 2024) with a large 128K vocabulary, this percentage even increases to 64% within the top-60 entries of the last logit. The results of 6 different LLMs on 13 downstream tasks demonstrate that this observation holds consistently across different models (Vicuna, Llama 3.1, and Qwen) and scales (ranging from 0.5B to 33B).

We further conduct experiments on Spec-Bench with Vicuna 7B to investigate the robustness of the predictive capability of the last logit across different tasks. We propose a minimal speculative

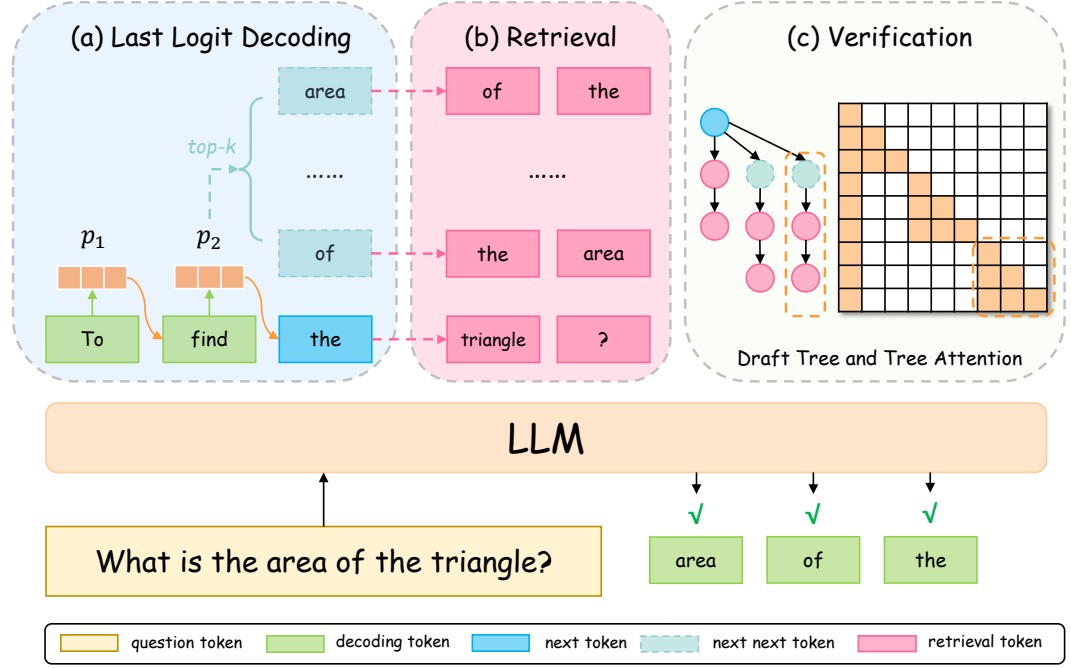

Figure 3: An overview of *LogitSpec*. At each decoding step, *LogitSpec* first utilizes the top-$k$ entries of the last logit as the speculation for the next next token. Then, *LogitSpec* retrieves relevant references for both the next token and the next next token. Finally, *LogitSpec* organizes the draft tokens into a draft tree and prepares a tree attention for parallel verification.

decoding algorithm with last logit, *last logit decoding* for this experiment, where we directly use the top-60 entries of the last logit to serve as the draft tokens. Each draft token is a guess at the next next token. If the target model accepts any one of the draft tokens, the others are dropped. Consequently, at each decoding step, the accepted length is either 2 or 1. As in 2(b), this simple SD method yields mean accepted tokens per decoding step (MAT) of over 1.5. Compared with other retrieval-based SD methods such as PLD (Saxena, 2023) and REST (He et al., 2024), which are highly affected by the task type, **the MAT of *last logit decoding* exhibits superior robustness to the task.** These results demonstrate that the ability of the last logit to predict the next next token widely exists in modern LLMs themselves, and strongly motivate us to improve retrieval-based SD methods with the last logit. We provide a detailed experimental setup for the motivation experiments in Appendix B.

## 4.2 *LogitSpec* DRAFTING

While *last logit decoding* has shown great simplicity in implementation and robustness to the downstream tasks, its theoretical upper bound of the speedup ratio is 2, as all the draft tokens are guesses for the next next token. In our experiments, the *last logit decoding* achieves an overall speedup ratio of $1.2\times \sim 1.4\times$, which hinders the real-world applications of *last logit decoding* due to the unsatisfactory speedup ratio. Therefore, to further improve the overall inference speedup, we propose to utilize the prediction ability of the last logit as **a guidance for retrieval**. Specifically, for a given input prefix $\mathbf{x} = (x_1, x_2, ..., x_i)$ and the last logit $p_i = \mathcal{M}(x_1, x_2, ..., x_i)$, we first sample the next token $x_{i+1} \sim p_i$. Then, we utilize the top-$k$ entries of the last logit (except the next token $x_{i+1}$) as the speculation to the next next token $\widetilde{x}_{i+2}$:

$$\mathcal{L} = \{\widetilde{x}_{i+2} \mid \widetilde{x}_{i+2} \in \text{TOP}_k(p_i) \wedge \widetilde{x}_{i+2} \neq x_{i+1}\}. \quad (4)$$

Then, we use the retrieval model $\mathcal{R}$ to retrieve reference for both the next token and next next tokens:

$$\mathcal{D} = \text{MATCH}(\mathcal{R}, \overbrace{(x_{i-m+2}, ..., x_i, x_{i+1})}^{m\text{-gram for next token}}) \cup \{\text{MATCH}(\mathcal{R}, \overbrace{(x_{i-m+3}, ..., x_i, x_{i+1}, \widetilde{x}_{i+2})}^{m\text{-gram for next next token}})\}_{\widetilde{x}_{i+2} \in \mathcal{L}} \quad (5)$$

Here, $m$-gram denotes the last $m$ tokens as the query to retrieve the matched reference. In our experiments, we retrieve draft tokens with $m = 3$ first. If no matched tokens are found, we decrease $m$ to 2 and so on. For simplicity, we only use the user-input prompt and decoded tokens to construct the retrieval model $\mathcal{R}$, that is to say, **our *LogitSpec* is query-independent, and will not be affected by other queries.** We give the detailed implementation of the retrieval process in Appendix D.1.

Taking the example in Figure 3, we first select the top-$k$ entries of the last logit as the speculation for the next next token. Then, we retrieve reference for both the next token "the" and the next next token "area", and return "of the". After that, we organize these draft tokens into a draft tree, where the next token serves as the root. Finally, the token sequence "area of the" is accepted by $\mathcal{M}$.

After the retrieval process, we employ a simple pruning strategy to control the number of draft tokens. Specifically, we do not prune the retrieval tokens for the next token. For each speculated next next token, we prune the retrieval tokens with a simple heuristic strategy: if the rank of the next next token is below 8, we preserve 4 tokens; if the rank of the next next token is below 32, we preserve 3 tokens; otherwise, we only preserve the speculated next next token itself. The retrieval process is terminated until the total number of draft tokens exceeds a specific draft tree capacity $K$. Furthermore, we explored the impact of various pruning strategies, with results detailed in Tables 13 of Appendix E.4.

### 4.3 *LogitSpec* VERIFICATION

After retrieving multiple draft token sequences $\mathcal{D}$, we organize these token sequences into a draft tree and prepare a tree attention for parallel verification.

Then, we prepare an attention mask to make each draft token sequence invisible to other sequences. Let $\mathrm{Tril}(l)$ denotes a causal mask with length $l$, and $m_j$ denotes the length of the $j$-th token sequence in $\mathcal{D}$, the attention mask is given by:

$$A_{\mathrm{draft}} = \mathrm{diag}(\mathrm{Tril}(m_1), \mathrm{Tril}(m_2), ..., \mathrm{Tril}(m_j)). \tag{6}$$

An illustration of this attention mask is shown in Figure 3(c) (note that the next token is visible to all sequences, corresponding to the first column). Both the preparation of the attention mask and the verification of the draft tree are consistent with previous works (Miao et al., 2024; Cai et al., 2024). We provide a pseudo code of the attention mask preparations in Appendix C.

## 5 EXPERIMENTS

### 5.1 EXPERIMENTAL SETUP

**Tasks and Datasets.** We conduct experiments on various text generation benchmarks to evaluate the effectiveness of *LogitSpec*. Specifically, we first evaluate the performance of *LogitSpec* on Spec-Bench (Xia et al., 2024). Spec-Bench is a widely used comprehensive benchmark that covers diverse application scenarios, including multi-turn conversation (MT), translation (Trans), summarization (Sum), question answering (QA), mathematical reasoning (Math), retrieval-augmented generation (RAG). Furthermore, we evaluate the performance of *LogitSpec* on HumanEval (Chen et al., 2021), GSM8K (Cobbe et al., 2021), CNN/DM (Nallapati et al., 2016), MATH (Hendrycks et al., 2021), AIME 24&25 (Mathematical Association of America, 2025) and LongBench (Bai et al., 2024). which are widely used text generation benchmarks in existing works.

**Baselines.** We compare *LogitSpec* to existing state-of-the-art *plug-and-play* SD methods: (a) Lookahead Decoding (Fu et al., 2024): utilizing the dropped draft tokens to serve as the retrieval model; (b) REST (He et al., 2024): employing an external knowledge base as the retrieval model; (c) PLD (Saxena, 2023): extracting n-grams from the user-input prompts and decoded tokens as the retrieval model; and (d) vanilla SpS (Leviathan et al., 2023; Chen et al., 2023): employing a well trained small-size model Vicuna-68M as the draft model. We use Vicuna series models (7B, 13B, 33B), Llama-2 series models (7B, 13B, 70B), Llama-3.1-Instruct-8B and Qwen-3-8B as backbones, and report results with baseline methods on Vicuna and Llama-2 as the main results, where the performance of baseline methods can be reproduced with their official implementations.

Table 1: Experimental results of *LogitSpec* on CNN/DM (Nallapati et al., 2016), GSM8K (Cobbe et al., 2021) and HumanEval (Chen et al., 2021) with **Vicuna**. We report the mean accepted tokens per decoding step (MAT) and overall speedup ratio. We **bold** the best results and underline the suboptimal results for each backbone model.

| Models | Method | CNN/DM | | GSM8K | | HumanEval | | Average | |
|---|---|---|---|---|---|---|---|---|---|
| | | MAT | Speedup | MAT | Speedup | MAT | Speedup | MAT | Speedup |
| **Vicuna 7B** | Lookahead | 1.54 | 1.28 | 1.92 | 1.64 | 1.79 | 1.57 | 1.75 | 1.50 |
| | REST | 1.64 | 1.19 | 1.55 | 1.13 | 1.96 | 1.51 | 1.72 | 1.28 |
| | PLD | 2.61 | 2.26 | 1.80 | 1.62 | 1.84 | 1.65 | 2.08 | 1.84 |
| | SpS | 2.34 | 1.58 | 2.08 | 1.49 | 2.56 | 1.80 | 2.33 | 1.62 |
| | *LogitSpec* | **3.28** | **2.61** | **2.69** | **2.20** | **2.62** | **2.24** | **2.86** | **2.35** |
| **Vicuna 13B** | Lookahead | 1.46 | 1.12 | 1.88 | 1.61 | 1.75 | 1.57 | 1.70 | 1.43 |
| | REST | 1.65 | 1.22 | 1.57 | 1.18 | 1.94 | 1.55 | 1.72 | 1.32 |
| | PLD | 2.34 | 1.85 | 1.76 | 1.61 | 1.98 | 1.77 | 2.03 | 1.74 |
| | SpS | 2.18 | 1.48 | 2.00 | 1.55 | 2.66 | 1.95 | 2.28 | 1.66 |
| | *LogitSpec* | **2.90** | **2.17** | **2.59** | **2.13** | **2.88** | **2.47** | **2.79** | **2.26** |
| **Vicuna 33B** | Lookahead | 1.50 | 1.18 | 1.89 | 1.44 | 1.66 | 1.33 | 1.68 | 1.32 |
| | REST | 1.65 | 1.41 | 1.57 | 1.32 | 1.99 | 1.64 | 1.74 | 1.46 |
| | PLD | 2.07 | 1.72 | 1.66 | 1.41 | 1.60 | 1.42 | 1.78 | 1.52 |
| | SpS | 2.01 | 1.57 | 1.97 | 1.54 | 2.27 | 1.80 | 2.08 | 1.64 |
| | *LogitSpec* | **2.66** | **2.01** | **2.46** | **1.93** | **2.41** | **1.95** | **2.51** | **1.96** |

**Implementation Details.** We follow the setting of Spec-Bench with Pytorch (Paszke et al., 2019) 2.6.0, transformers (Wolf et al., 2020) version of 4.37.1, and CUDA (NVIDIA et al., 2020) 12.4. The experiments where the model size is between 7B and 33B are conducted on single NVIDIA SXM A100 80G GPU, while for models with 70B size are conducted on 2 NVIDIA SXM A100 80G GPUs. The inference precision is `float16`. Following prior work, we conduct experiments with temperature 0 and batch size 1. The maximal generation lengths are 1024. More implementation details can be found in Appendix D. We use two widely used metrics for evaluation: mean accepted tokens per decoding step (denoted as MAT) and overall speedup ratio (denoted as Speedup).

## 5.2 MAIN RESULTS

We conduct experiments on various text generation benchmarks to demonstrate the effectiveness of *LogitSpec*. As in Tables 1 and 2, *LogitSpec* outperforms retrieval-based SD baselines by a large margin. We further present more in-depth analysis of *LogitSpec* including (i) more LLM backbones (Llama2 series, Llama-3.1-Instruct-8B and Qwen-3-8B), (ii) more benchmarks (MATH and AIME datasets), and (iii) long context benchmarks (the LongBench dataset) in Appendix E.

Specifically, we can get the following findings: (a) *LogitSpec* significantly improves the overall mean accepted tokens (MAT) of retrieval-based SD methods with an average MAT of $2.13 \sim 3.28$. Notably, *LogitSpec* achieves a speedup ratio of $2.61\times$ on CNN/DM with Vicuna 7B and $2.47\times$ on HumanEval with Vicuna 13B. In contrast, REST only achieves less than $1.5\times$ speedup of the vicuna model on the CNN/DM dataset. (b) Thanks to the robustness of the last logit, even in some sub-tasks that are difficult to retrieval-based SD methods, *LogitSpec* achieves a considerable speedup ratio. For example, in the Translation task in Spec-Bench, there exists nearly no available reference to serve as the draft tokens. Consequently, PLD and Lookahead achieve poor acceleration. Even equipped with a large external database ($\approx$ 12 GB), REST only achieves a speedup with $1.17\times \sim 1.31\times$, while our *LogitSpec* achieves a speedup with $1.38\times \sim 1.43\times$ without the external database, which also demonstrates the prediction ability on the next next token of the last logit. As *LogitSpec* is highly plug-and-play, it is also promising future work to employ an external database for further acceleration. (c) While *LogitSpec* and PLD are both retrieval-based SD methods that retrieve reference from the prompt, *LogitSpec* significantly outperforms PLD across all benchmarks and model sizes, highlighting the importance of "speculating the next next token", particularly in scenarios where the output is less duplicate to the prefix text. All these results validate that **the last logit can be used to predict the next next token and improve the retrieval performance.**

Table 2: Experimental results of *LogitSpec* on Spec-Bench (Xia et al., 2024) with **Vicuna**. We report the speedup ratio on each sub task, mean accepted tokens per decoding step (MAT) and overall speedup ratio. We **bold** the best results and underline the suboptimal results for each backbone model.

| Models | Method | MT | Trans | Sum | QA | Math | RAG | MAT | Speedup |
|---|---|---|---|---|---|---|---|---|---|
| **Vicuna 7B** | Lookahead | 1.40 | 1.14 | 1.19 | 1.24 | 1.55 | 1.09 | 1.66 | 1.27 |
| | REST | 1.63 | 1.31 | 1.36 | 1.66 | 1.21 | 1.73 | 1.82 | 1.48 |
| | PLD | 1.64 | 1.04 | 2.43 | 1.14 | 1.61 | 1.71 | 1.73 | 1.59 |
| | SpS | 1.66 | 1.13 | 1.62 | 1.49 | 1.47 | 1.55 | 2.28 | 1.49 |
| | *LogitSpec* | **1.92** | **1.39** | **2.68** | **1.70** | **2.22** | **1.87** | **2.44** | **2.01** |
| **Vicuna 13B** | Lookahead | 1.30 | 1.06 | 1.20 | 1.12 | 1.48 | 1.12 | 1.63 | 1.22 |
| | REST | 1.52 | 1.17 | 1.37 | **1.53** | 1.19 | 1.55 | 1.82 | 1.38 |
| | PLD | 1.47 | 1.02 | 2.19 | 1.03 | 1.57 | 1.71 | 1.68 | 1.48 |
| | SpS | 1.60 | 1.13 | 1.68 | 1.39 | 1.53 | 1.67 | 2.18 | 1.49 |
| | *LogitSpec* | **1.89** | **1.43** | **2.33** | 1.43 | **2.23** | **1.93** | **2.32** | **1.93** |
| **Vicuna 33B** | Lookahead | 1.32 | 1.08 | 1.20 | 1.06 | 1.54 | 1.15 | 1.61 | 1.24 |
| | REST | 1.63 | 1.27 | 1.45 | **1.61** | 1.30 | 1.61 | 1.80 | 1.48 |
| | PLD | 1.44 | 1.06 | 2.00 | 1.07 | 1.55 | 1.45 | 1.55 | 1.42 |
| | SpS | 1.75 | 1.28 | 1.76 | 1.53 | 1.69 | 1.68 | 2.01 | 1.61 |
| | *LogitSpec* | **1.77** | **1.38** | **2.15** | 1.37 | **2.00** | **1.69** | **2.13** | **1.76** |

Table 3: Ablation experiments of *LogitSpec* on Spec-Bench (Xia et al., 2024), CNN/DM (Nallapati et al., 2016), GSM8K (Cobbe et al., 2021) and HumanEval (Chen et al., 2021) with **Vicuna**. We report the ablation results with MAT reduction and Speedup reduction with down arrow ↓.

| Models | Method | Spec-Bench | | CNN/DM | | GSM8K | | HumanEval | |
|---|---|---|---|---|---|---|---|---|---|
| | | MAT | Speedup | MAT | Speedup | MAT | Speedup | MAT | Speedup |
| **Vicuna 7B** | *w/o last logit* | $1.72_{\downarrow.72}$ | $1.27_{\downarrow.74}$ | $2.66_{\downarrow.62}$ | $2.24_{\downarrow.37}$ | $1.81_{\downarrow.88}$ | $1.63_{\downarrow.57}$ | $1.89_{\downarrow.73}$ | $1.69_{\downarrow.55}$ |
| | *w/o retrieval* | $1.57_{\downarrow.87}$ | $1.24_{\downarrow.77}$ | $1.57_{\downarrow1.71}$ | $1.23_{\downarrow1.38}$ | $1.71_{\downarrow.98}$ | $1.39_{\downarrow.81}$ | $1.69_{\downarrow.93}$ | $1.39_{\downarrow.85}$ |
| | *LogitSpec* | **2.44** | **2.01** | **3.28** | **2.61** | **2.69** | **2.20** | **2.62** | **2.24** |
| **Vicuna 13B** | *w/o last logit* | $1.66_{\downarrow.66}$ | $1.23_{\downarrow.70}$ | $2.34_{\downarrow.56}$ | $1.81_{\downarrow.36}$ | $1.78_{\downarrow.81}$ | $1.61_{\downarrow.52}$ | $2.02_{\downarrow.86}$ | $1.75_{\downarrow.72}$ |
| | *w/o retrieval* | $1.59_{\downarrow.73}$ | $1.19_{\downarrow.74}$ | $1.58_{\downarrow1.32}$ | $1.22_{\downarrow.95}$ | $1.71_{\downarrow.88}$ | $1.45_{\downarrow.68}$ | $1.63_{\downarrow1.25}$ | $1.37_{\downarrow1.10}$ |
| | *LogitSpec* | **2.32** | **1.93** | **2.90** | **2.17** | **2.59** | **2.13** | **2.88** | **2.47** |
| **Vicuna 33B** | *w/o last logit* | $1.55_{\downarrow.58}$ | $1.22_{\downarrow.54}$ | $2.05_{\downarrow.61}$ | $1.72_{\downarrow.29}$ | $1.64_{\downarrow.82}$ | $1.39_{\downarrow.54}$ | $1.59_{\downarrow.82}$ | $1.40_{\downarrow.55}$ |
| | *w/o retrieval* | $1.61_{\downarrow.52}$ | $1.25_{\downarrow.51}$ | $1.62_{\downarrow1.04}$ | $1.26_{\downarrow.75}$ | $1.69_{\downarrow.77}$ | $1.33_{\downarrow.60}$ | $1.60_{\downarrow.81}$ | $1.34_{\downarrow.61}$ |
| | *LogitSpec* | **2.13** | **1.76** | **2.66** | **2.01** | **2.46** | **1.93** | **2.41** | **1.95** |

## 5.3 Ablation Study

To further investigate the components of *LogitSpec* and provide more insights, we conduct extensive ablation studies on the aforementioned 4 benchmarks in Table 3. Specifically, we mainly focus on two components of *LogitSpec*: *last logit* and *retrieval*. We denote *LogitSpec* without the last logit decoding and only using retrieval as *w/o last logit*, and *LogitSpec* without retrieval and only using last logit decoding as *w/o retrieval*. As shown in Table 3, the absence of any component results in a performance degradation of the entire framework.

Our findings are as follows. First, the absence of *retrieval* exhibits more importance to the final acceleration, which is consistent with the discussion in Section 4.2 that the theoretical upper bound of *last logit decoding* severely hinders its real-world application. Second, the absence of *retrieval* and the absence of *last logit* show different effects in different sub-tasks. For example, the absence of *retrieval* decreases MAT by 0.87 on Spec-Bench with Vicuna 7B, while it decreases MAT by 1.71 on CNN/DM. In contrast, the absence of *last logit* leads to a more consistent MAT degradation across different tasks, which further highlights the robustness of *last logit decoding*. Finally, these results demonstrate that combining *last logit* with *retrieval* improves the retrieval performance and overall speedup. We also conduct ablation studies on pruning strategies in the Appendix. E.4 and E.5. The experimental results suggest that varying the pruning strategy yields only minor differences.

## 5.4 CASE STUDY

**In-depth Running Time Analysis.** We conduct experiments to analyze the running time allocation of the whole decoding. Specifically, there are **five non-negligible components** in *LogitSpec*, including (a) *retrieving draft tokens*: the process of retrieving reference for the next token and the next next tokens; (b) *preparation*: preparing attention mask for the draft tokens; (c) *model forward*: conducting one-pass model forward; (d) *verification*: validating the draft tokens with speculative sampling; (e) *update*: necessary update of KV cache and retrieval model.

As shown in Figure 4, *model forward* occupies the majority of wall-clock time. Compared with vanilla AR decoding, the overhead introduced by *LogitSpec*, i.e., retrieving overhead, only takes 1.17% of the whole decoding process. It can be further alleviated by parallel techniques, as the retrieval process is independent. The process of *verification* and *update* takes 1.03% and 1.05% through the whole process, respectively, which brings negligible overhead as well.

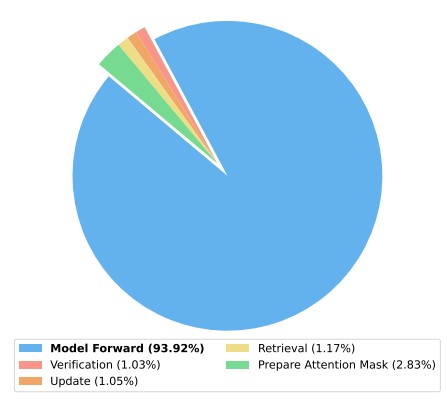

Figure 4: Running time breakdown of the whole decoding process on Spec-Bench with Vicuna 7B.

**Retrieval Performance.** As mentioned in Section 4.2, *LogitSpec* expands the retrieval range and improve the retrieval performance. We conduct a simple experiment to test the retrieval success rate, i.e., whether the model successfully retrieves the reference as draft tokens. As in Table 4, PLD cannot retrieve any matched tokens in more than 30% decoding steps, while *LogitSpec* retrieves matched reference in most decoding steps. These results further demonstrate the effectiveness of *LogitSpec*.

**Real-world examples.** We also provide a real-world example in Appendix. E.6 to illustrate how *next-next token speculation* allows *LogitSpec* to succeed where standard retrieval methods fail.

Table 4: Case study experiments of *LogitSpec* on Spec-Bench and HumanEval with **Vicuna**. We report the successful retrieval rate of each method. We report the relative improvements with ↑.

| Method | Spec-Bench | | | HumanEval | | |
|---|---|---|---|---|---|---|
| | Vicuna 7B | Vicuna 13B | Vicuna 33B | Vicuna 7B | Vicuna 13B | Vicuna 33B |
| PLD | 63.88 | 63.08 | 62.71 | 69.03 | 69.51 | 67.67 |
| *LogitSpec* | 97.76↑53.04% | 97.64↑54.79% | 97.93↑56.16% | 99.29↑43.84% | 99.31↑42.87% | 99.37↑46.84% |

## 6 CONCLUSION

In this paper, we empirically observe that the logit of the last token can predict **the next next token** with a relatively high accuracy without **any fine-tuning**. Based upon this observation, we propose a novel retrieval-based SD framework, namely *LogitSpec*, which utilizes the prediction ability of the last logit to effectively expand the retrieval range and find the most relevant reference as the draft tokens. *LogitSpec* does not require an additional draft model and is a fully **plug-and-play** method, which can be easily implemented and integrated into existing LLM frameworks. Extensive experiments demonstrate that *LogitSpec* can effectively improve the retrieval performance, leading to a $1.8\times \sim 2.6\times$ speedup across all the evaluation benchmarks.

**Limitations and Future Work.** While our *LogitSpec* is a fully plug-and-play SD framework, its real-world inference acceleration is less competitive. Our future works involve integrating *LogitSpec* into existing draft-model-based SD methods for further acceleration. Moreover, currently *LogitSpec* retrieves relevant reference from the prompt, which may incur lower speedup when the prompt is short. We consider integrating an external database to boost the retrieval model as a future work.

ETHICS STATEMENT

This work adheres to the ICLR Code of Ethics. Our study does not involve human subjects or personally identifiable information (PII), and we did not collect new sensitive data. All datasets are publicly available under their respective licenses, and third-party resources are credited. We report methods and results transparently, consider potential risks such as misuse or bias amplification, and do not recommend deployment in high-stakes settings without additional safety assessment.

REPRODUCIBILITY STATEMENT

We provide details to facilitate replication: dataset names and versions, preprocessing steps, model/configuration, training schedules, and evaluation protocols. All hyperparameters are listed in the appendix; complete scripts and configs are included in the anonymous supplementary materials. For theoretical or algorithmic components, assumptions and full proofs are provided in the appendix. These pointers collectively enable independent reproduction of our results.

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

## A  MORE DISCUSSIONS TO RELATED WORK

We give more discussions to existing works in both draft-model-based speculative decoding and draft-model-free speculative decoding. An intuitive comparison is shown in Table 5.

Table 5: Training and deployment comparison for different methods. Example training costs: EAGLE $8\times$ RTX 3090 for 1–2 days; HYDRA $8\times$ A100 for training; MEDUSA $1\times$ A100 for 5 hours.

| Methods | Training Cost | Additional Parameters | Lossless Quality? | Deployment Difficulty |
|---|---|---|---|---|
| EAGLE | High | AR Heads | ✓ | High |
| HYDRA | High | MLP Heads | ✗ | Moderate |
| MEDUSA | Moderate | MLP Heads | ✗ | Moderate |
| *LogitSpec* (Ours) | **None** | **None** | ✓ | **Plug-and-Play** |

**Draft-model-based speculative decoding.** Besides the discussions in Section 2, there are also many excellent works related to the draft-model-based speculative decoding. For example, HASS (Zhang et al., 2025) discovers the inconsistency of EAGLE between training and inference, and proposes a multi-step training framework to address this. CORAL (Weng et al., 2025) proposes a cross-step representation alignment to address this problem. Judge Decoding (Bachmann et al., 2025) recognizes the potential of accepting high-quality but refused draft tokens to further improve the acceleration. Gumiho (Li et al., 2025a) demonstrates that the initial draft token is more important and proposes a hybrid model to combine serial and parallel draft heads.

Besides these methods that focus on the training process, the process of verification also draws extensive interest, mainly focusing on adaptive draft length. SpecDec++ (Huang et al., 2025) formulates the verification process as a Markov decision process to adaptively determine the draft length. OPT-Tree (Wang et al., 2024) proposes a method to search for the optimal tree structure that maximizes the mathematical expectation of the acceptance length in each decoding step. AdaEAGLE (Zhang et al., 2024b) proposes a novel framework to explicitly model the draft tree structure for EAGLE series models. PEARL (Liu et al., 2025) pioneers this direction by serving the draft model and the target model in parallel to achieve a segmented adaptive draft length.

**Draft-model-free speculative decoding.** For layer sparsity, SPEED (Hooper et al., 2024) proposes a method to speculatively execute multiple future tokens in parallel with the current token using predicted values based on early-layer hidden states. FREE (Bae et al., 2023) proposes a shallow-deep module and a synchronized parallel decoding to improve the efficiency. EESD (Liu et al., 2024) proposes an early-exiting framework with a self-distillation method and leverages Thompson Sampling to regulate the generation processes. For retrieval-based SD, Token Recycling (Luo et al., 2024) proposes a method to utilize the dropped draft tokens and generate draft tokens via an adjacency matrix. Different from our method, Token Recycling is a **query-dependent** method that utilizes information from other queries, which may result in limitations when applied to real-world applications with complex and dynamic user inputs. SAM Decoding (Hu et al., 2024) utilizes a common text corpus and dynamic text sequence as retrieved sources and proposes a suffix automaton to efficiently obtain yields more accurate match positions.

## B  EXPERIMENTAL SETUP FOR THE MOTIVATION EXPERIMENTS

To investigate the prediction ability of the last logit, we conduct two motivation experiments to demonstrate the effectiveness and robustness of the last logit. For the effectiveness of the last logit, as shown in Figure 2(a), we conduct auto-regressive inference for the 6 models and record the logits for each decoded token. Then, for each decoded token $x_i$ which is sampled from the logit $p_{i-1}$, we investigate the rank of $x_i$ in $p_{i-2}$, i.e., the corresponding last logit, and visualize the statistics of the rank in Figure 2(a). For the robustness of the last logit, as shown in Figure 2(b), we conduct *last logit decoding* and investigate its mean accepted tokens per decoding step. Both experiments are conducted on a small subset of Spec-Bench, where we randomly sample 2 questions for each sub-category of MT, and 10 questions for other categories (Trans, Sum, QA, Math, RAG).

## C    PSEUDO CODE TO PREPARE ATTENTION INPUTS

We give a pseudo code to organize the retrieved multiple draft token sequences into a draft tree and prepare its attention mask. More detailed implementations can be found in our attached code.

```python
def prepare_attention_inputs(past_len, next_token, candidate_list,
    num_draft_tokens):
    'LogitSpec organizes draft tokens in a tree manner. Each sub-
        sequence corresponds to a local causal mask.'

    seq_len = num_draft_tokens + 1

    # organize the candidate list into a sequence
    draft_ids = [next_token] + [token for sub in candidate_list
        for token in sub]

    # prepare original position ids and attention mask
    position_ids = torch.zeros((1, seq_len), dtype=torch.long)
    causal_mask = torch.full((seq_len, past_len + seq_len),
        fill_value=0)
    causal_mask[:, :past_len+1] = 1

    # prepare causal mask
    idx = 1
    for sub_sequence in candidate_list:
        l = len(sub_sequence)
        sub_mask = torch.tril(torch.ones((l, l)))
        causal_mask[idx:idx+l, idx+past_len:idx+past_len+l] =
            sub_mask
        position_ids[0, idx:idx+l] = torch.arange(l) + 1
        idx += l

    position_ids += past_len
    return draft_ids, causal_mask, position_ids
```

## D    MORE IMPLEMENTATION DETAILS

### D.1    RETRIEVAL PROCESS

The retrieval process of retrieval-based SD methods will affect the overall acceleration. While the existing method PLD retrieves a reference with string matching, it is unaffordable for *LogitSpec* to retrieve in this way. Specifically, suppose the matching ngram size is $m$, the length of the whole token sequence is $n$, and string matching takes the time complexity of $O(mn)$. However, as *LogitSpec* needs to retrieve for all the next next tokens, the retrieval takes the time complexity of $O(kmn)$, where $k$ is the number of the next next tokens.

To address this issue, we propose a simple yet effective method to reduce the retrieval overhead by constructing a hash table. Specifically, we go through the token sequence, storing each key ngram with size 1 to $m$ and its value ngram (its following tokens). In this way, the retrieval overhead is reduced to $O(n + k)$ with an additional memory cost $O(mn)$. As the length of the token sequence is relatively small, the additional memory cost is negligible. Furthermore, we will update the retrieval model $\mathcal{R}$ with the decoded tokens in the same way during each decoding step.

### D.2    EVALUATION INSTRUCTIONS

In our experiments, we employ different instructions for different evaluation tasks. Specifically, for Spec-Bench, we use its standard instructions:

```
Prompt Templates for Spec-Bench

A chat between a curious user and an artificial intelligence
assistant.  The assistant gives helpful, detailed, and polite
answers to the user's questions.

USER: Question

ASSISTANT:
```

For CNN/DM, we prepend the instruction "Summarize: " to the instruction:

```
Prompt Templates for CNN/DM

A chat between a curious user and an artificial intelligence
assistant.  The assistant gives helpful, detailed, and polite
answers to the user's questions.

USER: Summarize:  Question

ASSISTANT:
```

For GSM8K, we follow the setting with (Liu et al., 2025) and use an 8-shot CoT for inference:

```
Prompt Templates for GSM8K

A chat between a curious user and an artificial intelligence
assistant.  The assistant gives helpful, detailed, and polite
answers to the user's questions.

USER: 8-shot CoT Q: Question

ASSISTANT:
```

For HumanEval, we add a simple instruction "Please help me to complete this code, just output your codes directly.":

```
Prompt Templates for HumanEval

A chat between a curious user and an artificial intelligence
assistant.  The assistant gives helpful, detailed, and polite
answers to the user's questions.

USER: Please help me to complete this code, just output your
codes directly.  Question

ASSISTANT:
```

### D.3 DATASET CONFIGURATIONS

In our experiments, we evaluate the effectiveness of our *LogitSpec* on 4 categories of text generation tasks, including Spec-Bench, CNN/DM, GSM8K, and HumanEval. For Spec-Bench and HumanEval, we use the full data for evaluation. For CNN/DM and GSM8K, we randomly sample 1000 questions for evaluation. The maximal generation length is set as 1024 across all the experiments.

## D.4 MODEL CONFIGURATIONS

In our experiments, all the models are deployed with precision `float16`. The `<|eos token|>` is set the same as the tokenizer's `<|eos token|>`. To effectively alleviate the overhead of rolling back the KV cache to the accepted draft tokens, we follow Medusa (Cai et al., 2024) to allocate continuous GPU memory for all the KV cache.

# E MORE IN-DEPTH ANALYSIS OF *LogitSpec*

## E.1 MORE RESULTS OF DIFFERENT LLM BACKBONES

Here, we provide more results for different LLM backbones to provide more insights into our *LogitSpec*, including Llama 2 chat series models, LLaMA-3.1-8B-Instruct, and Qwen3-8B.

We present the results of Llama 2 chat series models on Spec-Bench, CNN/DM, GSM8K, and HumanEval in Table 6 and Table 7. We still find that *LogitSpec* achieves the best acceleration results among all other baselines, which demonstrates the effectiveness of our *LogitSpec*.

We also present more results applying LLaMA-3.1-8B-Instruct and Qwen3-8B on Spec-Bench, CNN/DM, GSM8K, and HumanEval in Tables 8 and 9. We still observe that compared to standard LLM auto-regressive decoding, *LogitSpec* can consistently achieve around $2 \times$ inference acceleration for all datasets.

Table 6: Experimental results of *LogitSpec* on CNN/DM (Nallapati et al., 2016), GSM8K (Cobbe et al., 2021) and HumanEval (Chen et al., 2021) with **Llama-2**. We report the mean accepted tokens per decoding step (MAT) and overall speedup ratio. We **bold** the best results and underline the suboptimal results for each backbone model.

| Models | Method | CNN/DM | | GSM8K | | HumanEval | | Overall | |
|---|---|---|---|---|---|---|---|---|---|
| | | MAT | Speedup | MAT | Speedup | MAT | Speedup | MAT | Speedup |
| **Llama 2 7B** | Lookahead | 1.58 | 1.36 | 2.02 | 1.67 | 1.77 | 1.53 | 1.79 | 1.52 |
| | REST | 1.71 | 1.33 | 1.51 | 1.18 | 1.97 | 1.52 | 1.73 | 1.34 |
| | PLD | 1.89 | 1.73 | 3.32 | 2.98 | 1.57 | 1.41 | 2.26 | 2.04 |
| | SpS | 1.99 | 1.46 | 2.83 | 1.87 | 2.13 | 1.51 | 2.32 | 1.61 |
| | *LogitSpec* | **2.41** | **2.02** | **4.44** | **3.68** | **2.17** | **1.75** | **3.01** | **2.48** |
| **Llama 2 13B** | Lookahead | 1.56 | 1.18 | 2.08 | 1.52 | 1.84 | 1.66 | 1.83 | 1.45 |
| | REST | 1.71 | 1.34 | 1.53 | 1.26 | 1.96 | 1.63 | 1.73 | 1.41 |
| | PLD | 1.89 | 1.52 | 3.24 | 2.54 | 1.73 | 1.63 | 2.29 | 1.90 |
| | SpS | 1.95 | 1.34 | 2.87 | 1.85 | 2.33 | 1.76 | 2.38 | 1.65 |
| | *LogitSpec* | **2.43** | **2.03** | **4.31** | **3.24** | **2.38** | **2.10** | **3.04** | **2.46** |
| **Llama 2 70B** | Lookahead | 1.53 | 1.28 | 1.90 | 1.57 | 1.86 | 1.57 | 1.76 | 1.47 |
| | REST | 1.67 | 1.35 | 1.63 | 1.32 | 1.96 | 1.66 | 1.75 | 1.44 |
| | PLD | 1.98 | 1.74 | 1.63 | 1.46 | 1.62 | 1.49 | 1.74 | 1.56 |
| | SpS | 2.01 | 1.71 | 1.98 | 1.69 | 2.21 | 1.70 | 2.07 | 1.70 |
| | *LogitSpec* | **2.67** | **2.10** | **2.37** | **1.87** | **2.33** | **1.93** | **2.46** | **1.97** |

## E.2 MORE RESULTS OF DIFFERENT BENCHMARKS

We provide more results of our *LogitSpec* on complex math reasoning tasks including MATH (Hendrycks et al., 2021) and AIME 24 & 25 (Mathematical Association of America, 2025) datasets in Tables 11 and 10. For each of these benchmarks, we report the real-world speedup for each subset, the overall mean accepted tokens per decoding step (MAT), and the overall speedup. These expanded experimental results further corroborate *LogitSpec*'s effectiveness

Table 10: Results of *LogitSpec* on AIME datasets.

| Method | AIME24 | | AIME25 | |
|---|---|---|---|---|
| | MAT | Speedup | MAT | Speedup |
| Vanilla | 1.00 | 1.00 | 1.00 | 1.00 |
| *LogitSpec* | 3.41 | 3.25 | 3.76 | 3.33 |

Table 7: Experimental results of *LogitSpec* on Spec-Bench (Xia et al., 2024) with **Llama-2**. We report the speedup ratio on each sub task, mean accepted tokens per decoding step (MAT) and overall speedup ratio. We **bold** the best results and underline the suboptimal results for each backbone model.

| Models | Method | MT | Trans | Sum | QA | Math | RAG | MAT | Speedup |
|---|---|---|---|---|---|---|---|---|---|
| **Llama 2 7B** | Lookahead | 1.55 | 1.44 | 1.42 | 1.48 | 1.69 | 1.46 | 1.69 | 1.51 |
| | REST | 1.58 | 1.20 | 1.38 | 1.61 | 1.31 | 1.55 | 1.83 | 1.48 |
| | PLD | 1.46 | 1.34 | 1.89 | 1.23 | 1.65 | 1.58 | 1.59 | 1.49 |
| | SpS | 1.57 | 1.38 | 1.55 | 1.46 | 1.55 | 1.60 | 2.07 | 1.53 |
| | *LogitSpec* | **1.83** | **1.72** | **2.19** | **1.63** | **2.15** | **1.94** | **2.15** | **1.87** |
| **Llama 2 13B** | Lookahead | 1.39 | 1.36 | 1.21 | 1.39 | 1.69 | 1.25 | 1.68 | 1.37 |
| | REST | 1.53 | 1.14 | 1.30 | 1.51 | 1.23 | 1.45 | 1.85 | 1.42 |
| | PLD | 1.36 | 1.19 | 1.58 | 1.16 | 1.62 | 1.37 | 1.56 | 1.37 |
| | SpS | 1.52 | 1.26 | 1.43 | 1.42 | 1.54 | 1.50 | 2.03 | 1.48 |
| | *LogitSpec* | **1.73** | **1.57** | **1.86** | **1.53** | **2.14** | **1.73** | **2.12** | **1.74** |
| **Llama 2 70B** | Lookahead | 1.45 | 1.35 | 1.28 | 1.38 | 1.71 | 1.31 | 1.66 | 1.41 |
| | REST | 1.63 | 1.33 | 1.38 | **1.67** | 1.35 | 1.55 | 1.83 | 1.53 |
| | PLD | 1.34 | 1.32 | 1.76 | 1.18 | 1.63 | 1.47 | 1.51 | 1.39 |
| | SpS | 1.65 | 1.50 | 1.62 | 1.57 | 1.70 | 1.68 | 1.88 | 1.63 |
| | *LogitSpec* | **1.66** | **1.58** | **1.95** | 1.58 | **2.03** | **1.78** | **2.12** | **1.72** |

Table 8: Experimental results of *LogitSpec* on Spec-Bench (Xia et al., 2024) with **LLaMA-3.1-8B-Instruct and Qwen3-8B**. We report the speedup ratio on each sub task, mean accepted tokens per decoding step (MAT) and overall speedup ratio. We **bold** the best results for each backbone model.

| Model | Method | MT | Trans | Sum | QA | MATH | RAG | MAT | Speedup |
|---|---|---|---|---|---|---|---|---|---|
| Llama-3.1-8B-Instruct | Vanilla | 1.00 | 1.00 | 1.00 | 1.00 | 1.00 | 1.00 | 1.00 | 1.00 |
| Llama-3.1-8B-Instruct | *LogitSpec* | **1.89** | **1.67** | **1.94** | **1.68** | **2.01** | **1.77** | **2.11** | **1.88** |
| Qwen-3-8B | Vanilla | 1.00 | 1.00 | 1.00 | 1.00 | 1.00 | 1.00 | 1.00 | 1.00 |
| Qwen-3-8B | *LogitSpec* | **1.71** | **1.74** | **1.64** | **1.65** | **1.89** | **1.68** | **1.95** | **1.75** |

Table 9: Experimental results of *LogitSpec* on CNN/DM (Nallapati et al., 2016), GSM8K (Cobbe et al., 2021) and HumanEval (Chen et al., 2021) with **LLaMA-3.1-8B-Instruct and Qwen3-8B**. We report the mean accepted tokens per decoding step (MAT) and overall speedup ratio. We **bold** the best results for each backbone model.

| Model | Method | CNN/DM | | GSM8K | | Humaneval | | Overall | |
|---|---|---|---|---|---|---|---|---|---|
| | | MAT | Speedup | MAT | Speedup | MAT | Speedup | MAT | Speedup |
| Llama-3.1-8B-Instruct | Vanilla | 1.00 | 1.00 | 1.00 | 1.00 | 1.00 | 1.00 | 1.00 | 1.00 |
| Llama-3.1-8B-Instruct | *LogitSpec* | **2.04** | **1.85** | **2.18** | **1.95** | **2.63** | **2.31** | **2.28** | **2.04** |
| Qwen-3-8B | Vanilla | 1.00 | 1.00 | 1.00 | 1.00 | 1.00 | 1.00 | 1.00 | 1.00 |
| Qwen-3-8B | *LogitSpec* | **1.77** | **1.59** | **2.18** | **1.94** | **2.18** | **1.92** | **2.04** | **1.82** |

and applicability, yielding an overall MAT of 3.32 on MATH and 3.76 on AIME, with a corresponding overall speedup of 2.78x on MATH and 3.33x on AIME. This demonstrates *LogitSpec*'s robust performance even on challenging reasoning tasks.

### E.3 MORE RESULTS OF LONG-TEXT DATA SCENARIOS

We conduct a new set of experiments on the LongBench using Llama-3.1-8B-Instruct as the backbone model, randomly sampling 100 problems for evaluation. We observe that *LogitSpec* is highly effective in these long-context scenarios. As shown in Table 12, our method achieves an overall MAT of 3.09 and an overall speedup of 2.01. This strong performance is directly linked to the core mechanics of

Table 11: Experimental results of LogitSpec on MATH datasets with Llama 3.1-8B-Instruct as the backbone model.

| Method | Algebra | Probability | Geometry | Intermediate Algebra | Number Theory | Prealgebra | Precalculus | MAT | Speedup |
|--------|---------|-------------|----------|---------------------|---------------|------------|-------------|-----|---------|
| Vanilla | 1.00 | 1.00 | 1.00 | 1.00 | 1.00 | 1.00 | 1.00 | 1.00 | 1.00 |
| *LogitSpec* | **2.48** | **3.24** | **2.96** | **2.97** | **2.53** | **2.16** | **3.01** | **3.32** | **2.78** |

our method. *LogitSpec*'s retrieval model is built from the user-input prompt and previously decoded tokens. A longer context generally provides a richer database for this retrieval process. However, a large context also increases the chance of finding ambiguous or incorrect n-grams, which is precisely where *LogitSpec*'s "next next token speculation" offers a significant advantage. By using a more specific multi-token query, it effectively disambiguates the retrieval process, which is particularly crucial in a long context with many repetitive phrases. Furthermore, our retrieval implementation was designed for efficiency, using a hash table to ensure that the overhead remains negligible (around 1.17% of the total decoding time) even as the sequence length grows. Therefore, these new results confirm that *LogitSpec* is a robust and effective solution for accelerating inference in demanding long-context scenarios.

Table 12: Experimental results of LogitSpec on LongBench datasets with Llama 3.1-8B-Instruct as the backbone model.

| Method | qasper | multifieldqa | hotpotqa | wikimqa | report | qmsum | news | vcsum | trec | MAT | Speedup |
|--------|--------|--------------|----------|---------|--------|-------|------|-------|------|-----|---------|
| Vanilla | 1.00 | 1.00 | 1.00 | 1.00 | 1.00 | 1.00 | 1.00 | 1.00 | 1.00 | 1.00 | 1.00 |
| *LogitSpec* | **1.60** | **3.30** | **1.76** | **2.06** | **2.06** | **1.59** | **1.89** | **2.89** | **1.53** | **3.09** | **2.01** |

### E.4 MORE RESULTS OF DIFFERENT PRUNING STRATEGIES

In Section 4.2, we apply a heuristic pruning strategy to control the number of draft tokens. The role of the pruning algorithm is to effectively control the size of the resulting draft tree, keeping the decoding overhead low without sacrificing too many possibilities, which is a strategy to balance breadth and cost. To provide more insight into the pruning strategy, we provide a series of ablation studies on the pruning strategy in Table 13. Specifically, we consider two different pruning strategies:

(a) Strategy 1: A rank-based heuristic: if the rank is $<4$, we preserve 5 tokens; $<8$, 4 tokens; $<16$, 3 tokens; $<32$, 2 tokens; else 1 token.

(b) Strategy 2: A simple heuristic: preserving 4 tokens for all speculated next next tokens.

The results in Table 13 demonstrate that while different pruning strategies have some effect on the final performance, the overall performance of *LogitSpec* is quite robust. The speedup remains stable at 1.9x across these different approaches.

We also would like to clarify that the core contribution of our work is **next next token speculation** using the last logit to guide and enhance the efficiency and accuracy of retrieval-based speculative decoding. The primary purpose of the pruning algorithm is to serve as an auxiliary module for our core mechanism.

Table 13: Ablation study on pruning strategies. *LogitSpec* $_{with\ s1}$ and *LogitSpec* $_{with\ s2}$ denote *LogitSpec* with Strategy 1 and Strategy 2, respectively.

| Method | MT | Trans | Sum | QA | MATH | RAG | MAT | Speedup |
|--------|-----|-------|-----|-----|------|-----|-----|---------|
| *LogitSpec* $_{with\ s1}$ | **1.90** | **1.68** | **1.94** | 1.66 | **2.04** | **1.78** | **2.12** | **1.88** |
| *LogitSpec* $_{with\ s2}$ | 1.86 | 1.67 | 1.87 | **1.68** | 2.03 | 1.74 | 2.02 | 1.85 |
| ***LogitSpec* (ori)** | 1.89 | 1.67 | **1.94** | **1.68** | 2.01 | 1.77 | 2.11 | **1.88** |

E.5    MORE RESULTS OF ABLATION STUDY ON PRUNING HYPERPARAMETERS K

As mentioned in Section 4.2, we set the capacity of the draft tree to K=64. To provide further insight, we evaluate a range of $K$ values from 32 to 128 in Table 14. These results reveal a clear trade-off: as K increases, the MAT per step improves (from 2.03 to 2.30), since a larger tree offers more opportunities for token acceptance. However, the overall speedup ratio peaks at K=64 (1.92×) and subsequently declines. This is because verifying an overly large draft tree introduces significant computational overhead that ultimately negates the advantages of a higher acceptance rate.

Table 14: Ablation study on pruning hyperparameters $K$ using LLaMA-3.1-8B-Instruct as the backbone model.

| Method | MT | Trans | Sum | QA | MATH | RAG | MAT | Speedup |
|---|---|---|---|---|---|---|---|---|
| Vanilla | 1.00 | 1.00 | 1.00 | 1.00 | 1.00 | 1.00 | 1.00 | 1.00 |
| *LogitSpec* 32 | 1.79 | 1.62 | 1.82 | 1.56 | 1.93 | 1.67 | 1.95 | 1.79 |
| *LogitSpec* 48 | 1.86 | 1.66 | 1.90 | 1.67 | 2.00 | 1.73 | 2.04 | 1.85 |
| ***LogitSpec*** 64 | **1.89** | **1.67** | **1.94** | **1.68** | **2.01** | **1.77** | 2.11 | **1.88** |
| *LogitSpec* 80 | 1.85 | 1.64 | 1.86 | 1.67 | 1.84 | 1.74 | 2.14 | 1.83 |
| *LogitSpec* 96 | 1.81 | 1.65 | 1.83 | 1.66 | 1.84 | 1.73 | 2.17 | 1.80 |
| *LogitSpec* 112 | 1.79 | 1.61 | 1.77 | 1.64 | 1.83 | 1.69 | 2.21 | 1.77 |
| *LogitSpec* 128 | 1.74 | 1.62 | 1.73 | 1.58 | 1.81 | 1.62 | **2.22** | 1.73 |

E.6    STEP-BY-STEP ACCELERATION PROCESS OF *LogitSpec* FOR A REAL-WORLD EXAMPLE

To obtain more insights into *LogitSpec*, we provide the following real-world example to illustrate how "next next token speculation" allows *LogitSpec* to succeed where standard retrieval methods fail.

Taking the prefix of

```
Q: A pen costs as much as a pencil and eraser combined. A pencil
   costs $1.20 and an eraser costs $0.30. How much will 8 pens
   cost?
A: To find the cost of 8 pens, we first need to find the cost of
   one pen. A pencil costs $1.20 and an eraser costs $0.30. The
```

At the first step, *LogitSpec* will first generate the next token "combined" and then speculate the next next token as follows:

- .
- total
- cost
- combination
- pen
- eraser

Then, *LogitSpec* extends each draft token with retrieved n-grams and verifies these draft tokens:

- [×] . A pencil costs
- [×] total they had
- [✓] cost of one pen
- [×] combination (not retrieved matched n-grams)
- [×] pen.
- [×] eraser costs $

Finally, with the guidance of the last logit, we successfully accept 3 draft tokens, generate "the combined cost of one pen". In the next decoding step, *LogitSpec* accepts 3 draft tokens as well and generates "the combined cost of one pencil and one eraser". However, without the speculation of the last logit, only the first n-gram " A pencil costs" can be retrieved.

## F  LLM USAGE

We used a large language model (LLM)–based writing assistant solely for grammar and wording improvements on draft text. The LLM did not generate research ideas, claims, proofs, algorithms, code, figures, or analyses, and it did not have access to any non-public data. All edits suggested by the LLM were manually reviewed and either accepted or rewritten by the authors, who take full responsibility for the final content. The LLM is not an author of this paper.

