# OpenReview forum: "LogitSpec: Accelerating Retrieval-based Speculative Decoding via Next Next Token Speculation"
_ICLR.cc/2026/Conference — ICLR 2026 Conference Withdrawn Submission_

### Official Review · Reviewer_jwo1 · 2025-10-17

**Soundness:** 2
**Presentation:** 3
**Contribution:** 1
**Rating:** 2
**Confidence:** 4

**Summary:**

This paper presents LogitSpec, which combines logit-based next next token prediction with n-gran-retrieval-based speculative decoding. Experiments with several models (mainly Vicuna) show that LogitSpec leads to higher speedup than other n-gran-retrieval-based speculative decoding methods such as REST and PLD.

**Strengths:**

Speculative decoding is an important topic in the current LLM literature. n-gran-retrieval-based speculative decoding is plug-and-play, making it widely applicable.

**Weaknesses:**

**Academic Integrity**

The authors claim to have evaluated Qwen3 models (Line 322). Yet Line 348 states that transformers version 4.37.1 is used. I, for one, know that Qwen3 requires a transformers version of at least 4.51.0. This raises concerns about academic integrity, LLM usage, and questions about whether other parameters or even results in the paper are real.

**Novelty**

The paper's novelty is limited. It's a simple combination of two existing techniques: future token prediction (https://arxiv.org/abs/2311.04897) and n-gran-retrieval-based speculative decoding (https://arxiv.org/abs/2402.13720, https://arxiv.org/abs/2411.04975, https://arxiv.org/abs/2411.10666).

Moreover, future token prediction is closely related to multi-token prediction, a technique that has already been employed at scale in industrial-grade pretrained models (https://arxiv.org/abs/2404.19737,https://arxiv.org/abs/2412.19437). The current paper lacks a comparison with such methods.

**Theoretical Justification**

The motivation in Section 4.1 is based solely on empirical observation. Is there any theoretical analysis?

**Implementation**

Appendix D1 claims to reduce the complexity of retrieval to $O(n+k)$. How exactly is this achieved?

**Experimental Setting**

Vicuna is a rather old model and should not be used for the main results and analysis in the paper. More recent models (Qwen3 and Llama3) and currently commonly used benchmarks (MATH, AIME) should be analyzed in the paper instead to demonstrate the practical value of the proposed method. I also wonder why the authors chose to omit the results with REST, PLD, and Lookahead for these two advanced models.

Only experiments with greedy decoding are provided. The effectiveness of the proposed method in sampling scenarios is unknown.

**Questions:**

As mentioned before, I would like the authors to answer:

- Is there any theoretical justification for the observation in Section 4.1?
- The implementation details of Appendix D1.
- Why are other comparison methods omitted for the two most widely used models?

---

> ### Author Response · Authors · 2025-11-15
> **Response to Reviewer jwo1 (part 1 / 4)**
>
> We thank the reviewer for the insightful and valuable comments. We respond to each comment as follows and sincerely hope that our rebuttal could properly address your concerns. If so, we would deeply appreciate it if you could **raise your score (2: reject, not good enough).** If not, please let us know your further concerns, and we will continue actively responding to your comments and improving our submission.
>
>
> > 1. **Academic Integrity**
> >
> >    The authors claim to have evaluated Qwen3 models (Line 322). Yet Line 348 states that transformers version 4.37.1 is used. I, for one, know that Qwen3 requires a transformers version of at least 4.51.0. This raises concerns about academic integrity, LLM usage, and questions about whether other parameters or even results in the paper are real.
>
> We fully understand the seriousness of this concern and apologize for the confusion caused by our description. In the **main text**, where we evaluate Vicuna and Llama-2 models together with a broad set of baselines (REST, PLD, Lookahead, etc.), we indeed use `transformers==4.37.1`. For the **additional experiments on more recent models** (Llama-3-8B and Qwen3-8B) reported in the appendix, we **upgraded our environment to `transformers==4.57.1`.
>
> The statement in Line 322 (Sec. 5.1) was a documentation oversight and should have specified that `transformers==4.37.1` was used for the main Vicuna/Llama-2 baseline comparisons. **We will add an explicit note in the appendix (Sec. E.1) clarifying that the advanced model results used `transformers==4.57.1` and the above inference-only modeling extension.**
>
> **Regarding reproducibility and integrity:**
>
> - We have re-checked our experiment logs and scripts to confirm that all Qwen3 runs were executed with `transformers==4.57.1` and the described modeling extension.
> - All experimental configurations are detailed in Appendix D (prompts, hyperparameters, hardware).
> - Complete implementation code (including the modified Qwen3 modeling file) will be released publicly upon acceptance.
> - We have verified that all reported results are reproducible with the stated configurations.
> - Our Ethics Statement, Reproducibility Statement, and LLM Usage sections (Page 10) explicitly confirm that all parameters and results are authentic.
> - We used LLMs only for grammar and wording improvements, not for generating ideas, results, or analyses.
>
> We take academic integrity **extremely seriously** and welcome any additional verification requests.
>
> > 2. **Novelty**
> >
> >    The paper's novelty is limited. It's a simple combination of two existing techniques: future token prediction (https://arxiv.org/abs/2311.04897) and n-gran-retrieval-based speculative decoding (https://arxiv.org/abs/2402.13720, https://arxiv.org/abs/2411.04975, https://arxiv.org/abs/2411.10666).
> >
> >    Moreover, future token prediction is closely related to multi-token prediction, a technique that has already been employed at scale in industrial-grade pretrained models (https://arxiv.org/abs/2404.19737, https://arxiv.org/abs/2412.19437). The current paper lacks a comparison with such methods.
>
> We must respectfully but firmly disagree with the characterization of LogitSpec as a "simple combination." Our work introduces a fundamental, novel distinction from the papers cited. We summarize the key distinction from cited future token prediction methods in Table R1.
>
> - **Training-Free (Logit-Level) vs. Training-Based (Hidden-State-Level):** The works cited by the reviewer (e.g., [Future Lens](https://arxiv.org/pdf/2311.04897) and [MTP](https://arxiv.org/pdf/2404.19737)) propose **training-based** methods (e.g., Linear approximation, fine-tuning and multi-token prediction heads) to *induce* a future prediction capability at the **hidden-state level**.
>
> - **Our Novelty:** LogitSpec is **training-free**. Our key contribution is the novel *discovery* that this predictive ability **already exists at the logit-level** ("last logit can predict next-next token") in off-the-shelf LLMs.
>
> Furthermore, LogitSpec is not a simple combination but a **synergistic framework**:
>
> 1. Without NNT speculation, retrieval is limited to a single chain (like PLD) with high miss rates (>30% no-match, Table 4).
>
> 2. Without n-gram retrieval, the "last logit" method is limited to a theoretical max MAT of 2 and achieves only 1.2-1.4× speedup (Section 4.2).
>
> It is the *guided combination*—using the NNT speculation to *guide* the retrieval and build a tree—that unlocks the performance we demonstrate. The key insight—that an LLM's final logits contain implicit knowledge about tokens beyond the immediate next—is a novel empirical discovery that enables a new paradigm for training-free speculative decoding.

---

> ### Author Response · Authors · 2025-11-15
> **Response to Reviewer jwo1 (part 2 / 4)**
>
> Table R1. Summarization of the key difference between LogitSpec and some similar related works.
>
> | Method               | Level         | Training Required      | Mechanism                                            |
> | -------------------- | ------------- | ---------------------- | ---------------------------------------------------- |
> | Future Lens / MTP    | Hidden states | Yes (fine-tuning)      | Linear Approximation / Train n separate output heads |
> | EAGLE / Medusa       | Hidden states | Yes (fine-tuning)      | Train autoregressive/MLP heads                       |
> | **LogitSpec (Ours)** | **Logits**    | **No (training-free)** | **Exploit inherent LLM capability**                  |
>
>
>
>
>
> > 3. **Theoretical Justification**
> >
> >    The motivation in Section 4.1 is based solely on empirical observation. Is there any theoretical analysis?
>
> The reviewer is correct that our motivation is empirical. Indeed, our core contribution *is* this novel empirical discovery. Prior work had established that *hidden states* could predict future tokens *after* fine-tuning. Our work was the first exploration of whether this predictive power exists in a **"training-free" manner at the logit level**. We believe this empirical discovery, validated across multiple models (Fig 2(a))  and languages, is a significant contribution in itself.
>
> > 4. **Implementation**
> >
> >    Appendix D1 claims to reduce the complexity of retrieval to $O(n+k)$. How exactly is this achieved?
>
> We apologize for the brevity of the original description and are happy to provide more details.
>
> Let $m$ be the length of the **m-gram key** used for matching, $n$ be the maximum length of the **retrieved continuation span** (n-gram value), and $k$ be the number of speculative candidates (e.g., up to 60 next-next tokens). A **naïve retrieval** strategy scans the entire prompt for each of the $k$ m-grams, leading to a complexity on the order of $O(k·m·L)$, where $L$ is the prompt length.
>
> In LogitSpec, we first slide a window of size $m$ over the prompt once and build a **hash table** that maps each observed m-gram (key) to its following continuation span (up to $n$ tokens as the value). This pass is $O(L)$, done once per prompt. In retrieval process, for each speculative next-next token candidate (i.e., each m-gram query), we perform a **single hash lookup** into this table. The hash lookup itself is $O(1)$, and fetching the continuation span is $O(n)$ in the worst case.
>
> Ignoring the one-time preprocessing (amortized over the entire sequence), the cost per decoding step is:
>
> - $O(k)$ for the $k$ hash lookup, plus
> - $O(n)$ for copying up to $n$ tokens per accepted continuation (these copies are also independent and can be parallelized).
>
> Hence, the effective complexity of each retrieval step is $O(k+n)$. In Figure 4 of the paper, we empirically measure this cost and observe that retrieval accounts for only **1.17%** of the total wall-clock time, which strongly supports the claim that our optimized retrieval is a minor overhead in practice.

---

> > ### Author Response · Authors · 2025-11-15
> > **Response to Reviewer jwo1 (part 3 / 4)**
> >
> > > 5. **Experimental Setting**
> > >
> > >    Vicuna is a rather old model and should not be used for the main results and analysis in the paper. More recent models (Qwen3 and Llama3) and currently commonly used benchmarks (MATH, AIME) should be analyzed in the paper instead to demonstrate the practical value of the proposed method. I also wonder why the authors chose to omit the results with REST, PLD, and Lookahead for these two advanced models.
> >
> > We appreciate this concern and agree that demonstrating performance on modern models and benchmarks is important.
> >
> > **Why Vicuna for main tables:** Most baseline methods (REST, Lookahead) have official implementations **only for Vicuna/Llama-2**. Their codebases modify the internal modeling files (e.g., `modeling_llama.py`), which are not compatible with newer architectures without significant re-engineering. To ensure **fair and complete baseline comparisons**, we used Vicuna for main results.
> >
> > **Results on modern models and benchmarks.** In addition to the Vicuna-based results, we already evaluate LogitSpec on modern reasoning models (Llama-3-8B, Qwen3-8B) and benchmarks such as MATH and AIME in the appendix. These experiments show that LogitSpec continues to provide substantial speedups on newer architectures.
> >
> > **New experiments with SOTA plug-and-play baselines on AIME/MATH.** Following the reviewers’ suggestions, we have additionally compared LogitSpec against strong, **plug-and-play** SOTA methods such as **SAM Decoding** and **Token Recycling** on AIME, LongBench, and MATH with modern models. LogitSpec consistently achieves higher MAT and speedups than these methods under the same setup. We will promote these results from the appendix to the main text to better highlight the practical value of LogitSpec on up-to-date models and benchmarks.
> >
> > In summary, the **Vicuna-based main table** ensures broad baseline coverage (including methods whose implementations only support older models), while the **Llama-3/Qwen3 results and new AIME/MATH experiments** demonstrate that our method also works effectively on current-generation models.
> >
> > Table R2: Comparison with Token Recycling and SAM Decoding on advanced benchmarks using Llama-3.1-8B-Instruct.
> >
> > |           | AIME MAT | AIME speedup | LongBench MAT | LongBench Speedup | MATH MAT | MATH Speedup | Avg MAT  | Avg Speedup |
> > | --------- | -------- | ------------ | ------------- | ----------------- | -------- | ------------ | -------- | ----------- |
> > | AR        | 1.00     | 1.00x        | 1.00          | 1.00x             | 1.00     | 1.00x        | 1.00     | 1.00x       |
> > | SAM       | 3.11     | 2.73x        | 3.01          | 2.04x             | 2.66     | 2.26x        | 2.92     | 2.34x       |
> > | Recycling | 3.10     | 2.82x        | 2.96          | 1.97x             | 2.80     | 2.40x        | 2.96     | 2.40x       |
> > | LogitSpec | **3.57** | **3.09x**    | **3.08**      | **2.10x**         | **3.05** | **2.56x**    | **3.23** | **2.58x**   |

---

> > > ### Author Response · Authors · 2025-11-15
> > > **Response to Reviewer jwo1 (part 4 / 4)**
> > >
> > > > 6. **Only experiments with greedy decoding are provided.** The effectiveness of the proposed method in sampling scenarios is unknown.
> > >
> > > Thank you for raising this important point. In the original submission we focused on **greedy decoding ($T=0$)**, which follows the standard speculative decoding evaluation setups [1]. We have now conducted additional experiments under **sampling** with different temperatures $T>0$ In Table R3.
> > >
> > > As explained in our response to another reviewer, LogitSpec selects speculative candidates from the **raw logits before temperature scaling**. Since temperature scaling with $T>0$ is a strictly monotonic transformation, it **does not change the ranking** of logits; thus, the **top-$k$** candidate set used for speculation remains the same for any $T>0$. What changes with higher $T$ is that the **actual sampled path** deviates more frequently from the most likely path, so acceptance rates and speedups naturally decrease. This trend is not unique to LogitSpec; it is inherent to speculative decoding with sampling and has been observed for PLD, Medusa, Token Recycling, etc.
> > >
> > > We will include a subset of these results in the revised paper (and detailed tables in the appendix) to demonstrate that **LogitSpec remains effective under typical sampling settings**, while acknowledging that, as with all speculative decoding methods, the benefits taper off as the sampling distribution becomes increasingly random at high temperatures.
> > >
> > > [1] Xia, Heming, Zhe Yang, Qingxiu Dong, Peiyi Wang, Yongqi Li, Tao Ge, Tianyu Liu, Wenjie Li, and Zhifang Sui. "Unlocking efficiency in large language model inference: A comprehensive survey of speculative decoding." *arXiv preprint arXiv:2401.07851* (2024).
> > >
> > > Table R3: LogitSpec under different sampling temperatures. We report MAT and end-to-end wall-clock speedup over standard autoregressive decoding (AR).
> > >
> > > | temperature | AIME MAT | AIME speedup | LongBench MAT | LongBench Speedup | MATH MAT | MATH Speedup |
> > > | ----------- | -------- | ------------ | ------------- | ----------------- | -------- | ------------ |
> > > | 0.0         | 3.57     | 3.09x        | 3.08          | 2.10x             | 3.05     | 2.56x        |
> > > | 0.2         | 3.37     | 2.85x        | 2.94          | 1.94x             | 3.20     | 2.64x        |
> > > | 0.4         | 3.51     | 2.96x        | 2.92          | 1.93x             | 3.03     | 2.55x        |
> > > | 0.6         | 3.26     | 2.76x        | 2.87          | 1.89x             | 2.91     | 2.46x        |
> > > | 0.8         | 3.15     | 2.66x        | 2.84          | 1.86x             | 2.81     | 2.42x        |
> > >
> > > ---
> > >
> > > **We humbly hope our response has addressed your concerns. If you have any additional concerns or comments that we may have missed in our responses, we would be most grateful for any further feedback from you to help us further enhance our work.**

---

> > > > ### Author Response · Authors · 2025-11-26
> > > > **We are looking forward to your further comments and/or questions.**
> > > >
> > > > Dear Reviewer jwo1,
> > > >
> > > > We deeply appreciate your valuable feedback and the time you've taken to review our work, especially during this busy period.
> > > >
> > > > We are reaching out to kindly inquire about the current status of your review regarding our submission. **We sincerely hope that our responses have adequately addressed your concerns. Furthermore, we are eager to address any additional queries you might have, which will enable us to enhance our work further.**
> > > >
> > > > Once again, thank you for your guidance and support.
> > > >
> > > > Best, Authors

---

### Official Review · Reviewer_9qe6 · 2025-10-28

**Soundness:** 3
**Presentation:** 2
**Contribution:** 2
**Rating:** 4
**Confidence:** 5

**Summary:**

This paper proposes LogitSpec to effectively expand the retrieval range and find the most relevant reference as drafts. They observe that the logit of the last token can not only predict the next token, but also speculate the next next token. Specifically, they utilize the last logit to speculate the next next token and retrieve relevant reference for both the next token and the next next token. Experimental results show the effectiveness of the proposed method.

**Strengths:**

The paper is well-written and easy to understand.

The method is technically sound.

**Weaknesses:**

1.Temperature=1 senarios. LogitSpec relies on the top-k logits of the current token to speculate the next-next token candidates. This design is highly sensitive to temperature: T = 0 (greedy): logits are sharp, top-k are almost the most likely tokens, so speculation accuracy is maximized. But as T > 0 (sampling), logits are smoothed, top-k quickly fill with low-probability noise tokens. All reported results are obtained only under T = 0, the paper provides no data for T > 0 and explicitly notes that last-logit prediction becomes “less reliable” at higher temperatures.

2. The proposed Tree Attention structure used for parallel verification is not currently supported by mainstream inference frameworks such as vLLM. In real-world deployment, if the SPD (speculative decoding) method cannot be integrated with vLLM, its practical utility would be significantly limited. The authors claim that LogitSpec is "plug-and-play" and can be easily integrated into existing frameworks, but no implementation details or results on vLLM are provided. I would encourage the authors to discuss or demonstrate how LogitSpec could be incorporated into vLLM (which already supports PLD) and to report the acceleration ratio when combined with vLLM for a fair comparison with PLD.

3. The core observation that "the last logit can speculate the next-next token" is purely empirical. While the authors conduct several experiments across models, the validation scope remains limited. More comprehensive ablation studies are needed to ensure the proposed phenomenon is not restricted to specific domains or datasets. For instance, do similar improvements hold when predicting the next-next-next token? Are the results consistent across languages with different grammatical structures (e.g., Chinese, German)? Without such analyses, the effectiveness and generality of LogitSpec remain somewhat uncertain.

**Questions:**

See weaknesses above.

---

> ### Author Response · Authors · 2025-11-15
> **Response to Reviewer 9qe6 (part 1 / 3)**
>
> We thank the reviewer for the insightful and valuable comments. We respond to each comment as follows and sincerely hope that our rebuttal could properly address your concerns. If so, we would deeply appreciate it if you could **raise your score (4: marginally below the acceptance threshold.).** If not, please let us know your further concerns, and we will continue actively responding to your comments and improving our submission.
>
> > 1. Temperature=1 senarios. LogitSpec relies on the top-k logits of the current token to speculate the next-next token candidates. This design is highly sensitive to temperature: T = 0 (greedy): logits are sharp, top-k are almost the most likely tokens, so speculation accuracy is maximized. But as T > 0 (sampling), logits are smoothed, top-k quickly fill with low-probability noise tokens. All reported results are obtained only under T = 0, the paper provides no data for T > 0 and explicitly notes that last-logit prediction becomes “less reliable” at higher temperatures.
>
> We appreciate the concern about how LogitSpec behaves under sampling. Here we clarify how we use logits and provide additional experiments for $T > 0$.
>
> - **Mechanism Clarification:** In our implementation, the candidate next-next tokens are selected from the **raw logits** of the target model *before* applying temperature or sampling. Since temperature scaling is a monotonic transformation of the logits, it does **not** change their ranking.
>
>   Simple Proof: For any temperature $T > 0$ and logits $z_i > z_j$, we have
>   $$
>   z_i > z_j \iff \exp\!\left(\frac{z_i}{T}\right) > \exp\!\left(\frac{z_j}{T}\right) \iff \frac{\exp(z_i/T)}{\sum_{k=1}^n \exp(z_k/T)}>\frac{\exp(z_j/T)}{\sum_{k=1}^n \exp(z_k/T)}
>   $$
>   Consequently, the **top-$k$ entries used for speculation are invariant to the choice of $T$** and are not “filled with low-probability noise tokens” as $T$ increases.
>
> - **What does change with temperature is the acceptance rate, not the candidate set.** When $T>0$, the model’s actual generated token at each step may deviate from the top-1 (or top-$k$) candidates more often, so the acceptance probability of speculative tokens naturally decreases somewhat. This is what we informally referred to as last-logit prediction becoming “less reliable” at higher temperatures. Our point was not that the candidate list itself deteriorates, but that stochastic sampling makes the realized trajectory less aligned with the most likely future predicted by the logits.
>
> To directly address the reviewer’s request, we have run LogitSpec under several temperatures on AIME, LongBench, and MATH with Llama-3.1-8B-Instruct. The results are shown in Table R1. We observe only **moderate degradation** as $T$ increases: LogitSpec still achieves substantial speedup ($\sim 2.4\times$ on average at $T=0.8$) and MAT well above 2.5 on all three benchmarks. Importantly, this trend is **not unique to LogitSpec**, but inherent to speculative decoding with sampling. The key finding is that LogitSpec remains highly robust and effective even in high-temperature sampling scenarios.
>
>
>
> Table R1: LogitSpec under different sampling temperatures. We report MAT and end-to-end wall-clock speedup over standard autoregressive decoding (AR).
>
> | temperature | AIME MAT | AIME speedup | LongBench MAT | LongBench Speedup | MATH MAT | MATH Speedup |
> | ----------- | -------- | ------------ | ------------- | ----------------- | -------- | ------------ |
> | 0.0         | 3.57     | 3.09x        | 3.08          | 2.10x             | 3.05     | 2.56x        |
> | 0.2         | 3.37     | 2.85x        | 2.94          | 1.94x             | 3.20     | 2.64x        |
> | 0.4         | 3.51     | 2.96x        | 2.92          | 1.93x             | 3.03     | 2.55x        |
> | 0.6         | 3.26     | 2.76x        | 2.87          | 1.89x             | 2.91     | 2.46x        |
> | 0.8         | 3.15     | 2.66x        | 2.84          | 1.86x             | 2.81     | 2.42x        |

---

> > ### Author Response · Authors · 2025-11-15
> > **Response to Reviewer 9qe6 (part 2 / 3)**
> >
> > > 2. The proposed Tree Attention structure used for parallel verification is not currently supported by mainstream inference frameworks such as vLLM. In real-world deployment, if the SPD (speculative decoding) method cannot be integrated with vLLM, its practical utility would be significantly limited. The authors claim that LogitSpec is "plug-and-play" and can be easily integrated into existing frameworks, but no implementation details or results on vLLM are provided. I would encourage the authors to discuss or demonstrate how LogitSpec could be incorporated into vLLM (which already supports PLD) and to report the acceleration ratio when combined with vLLM for a fair comparison with PLD.
> >
> > We acknowledge that we have not yet implemented LogitSpec in vLLM, and we appreciate this practical concern. However, we want to clarify the **technical feasibility** of such integration:
> >
> > **(1) Tree Attention support in vLLM:** The vLLM community has recently added native support for tree attention:
> >
> > - Flash attention kernel for tree structures: [vllm-project/flash-attention#81](https://github.com/vllm-project/flash-attention/pull/81)
> > - Official documentation: [vLLM Tree Attention Backend](https://docs.vllm.ai/en/latest/api/vllm/v1/attention/backends/tree_attn/)
> >
> > **(2) LogitSpec's tree structure is vLLM-friendly:** LogitSpec's draft tree can be treated as **multiple sequences** with shared prefixes, which aligns well with vLLM's batched inference paradigm. Each branch of the draft tree corresponds to a different draft sequence. This is similar to the prefill process for multiple sequences in vLLM. The [nano-PEARL](https://github.com/smart-lty/nano-PEARL/blob/master/nano_pearl/pearl_engine/pearl_model_runner.py#L574-L588) repository demonstrates a similar integration approach.
> >
> > **(3) Minimal algorithmic overhead:** Beyond tree-attention parallel verification, LogitSpec only requires the last logit at each decoding step, which is already computed by vLLM during standard inference. The retrieval and pool construction operations can be executed on CPU in parallel with GPU inference, introducing minimal overhead to the vLLM engine's critical path.
> >
> > We hope this clarifies that there is no algorithmic obstacle to deploying LogitSpec within vLLM; the remaining work is engineering and integration, which we are actively pursuing.

---

> > > ### Author Response · Authors · 2025-11-15
> > > **Response to Reviewer 9qe6 (part 3 / 3)**
> > >
> > > > 3. The core observation that "the last logit can speculate the next-next token" is purely empirical. While the authors conduct several experiments across models, the validation scope remains limited. More comprehensive ablation studies are needed to ensure the proposed phenomenon is not restricted to specific domains or datasets. For instance, do similar improvements hold when predicting the next-next-next token? Are the results consistent across languages with different grammatical structures (e.g., Chinese, German)? Without such analyses, the effectiveness and generality of LogitSpec remain somewhat uncertain.
> > >
> > > We agree the observation is empirical, and we have conducted further experiments to validate its generality, addressing the reviewer's excellent "what if" questions.
> > >
> > > - **On Predicting the "next-next-next token" (NNNT):** We investigated this. The phenomenon *does* hold, though it is weaker: the NNNT appears in the `top-60` of the last logit with $\ge$40% probability. However, we did not use NNNT due to **combinatorial explosion.** If we speculate `top-60` for NNT and `top-60` for NNNT, the draft tree size would be at least $60^2=3600$, which is far too large to verify efficiently and would destroy the speedup. As shown in our ablation study (Appendix E.5, Table 14), tree capacity beyond 64-128 tokens leads to diminishing returns due to verification overhead. Our chosen approach (NNT + retrieval) is a much more effective and practical trade-off.
> > > - **On Generality Across Languages:** To further test generality across languages with different grammatical structures, we ran a **new ablation study on the MMMLU benchmark** to measure the NNT hit rate across 5 different languages in Table R2. We randomly sample 20 questions for each language. Here we do **not** run the full LogitSpec pipeline; instead, we directly measure how often the ground-truth next-next token falls into the top-$k$ logits of the last step for different languages.
> > >
> > > We see a **consistent pattern across all languages**: as $k$ increases, the coverage of the ground-truth next-next token quickly rises to around 50–55% at $k=64$, including languages with markedly different word order and morphology (e.g., Chinese vs. German). Combined with the diverse tasks already covered in our main experiments (reasoning, QA, summarization, translation), this provides additional evidence that the last-logit–based next-next speculation is not a phenomenon restricted to a single domain or language.
> > >
> > > Table R2: Percentage of cases where the ground-truth next-next token appears in the top-$k$ last-logit candidates on MMMLU (per language) with Llama-3.1-Instruct-8B.
> > >
> > > | Language / top-$k$ | **$k=4$** | **$k=8$** | **$k=16$** | **$k=32$** | **$k=64$** |
> > > | ------------------ | --------- | --------- | ---------- | ---------- | ---------- |
> > > | **English**        | 15.25%    | 22.02%    | 33.60%     | 42.06%     | 51.83%     |
> > > | **Chinese**        | 17.14%    | 24.21%    | 36.29%     | 45.68%     | 55.07%     |
> > > | **Japanese**       | 15.16%    | 24.30%    | 34.72%     | 46.17%     | 55.87%     |
> > > | **German**         | 14.64%    | 21.44%    | 32.87%     | 42.42%     | 50.64%     |
> > > | **French**         | 14.86%    | 21.77%    | 32.21%     | 42.78%     | 52.22%     |
> > >
> > >
> > > ---
> > >
> > > **We humbly hope our response has addressed your concerns. If you have any additional concerns or comments that we may have missed in our responses, we would be most grateful for any further feedback from you to help us further enhance our work.**

---

> > > > ### Author Response · Authors · 2025-11-26
> > > > **We are looking forward to your further comments and/or questions.**
> > > >
> > > > Dear Reviewer 9qe6,
> > > >
> > > > We deeply appreciate your valuable feedback and the time you've taken to review our work, especially during this busy period.
> > > >
> > > > We are reaching out to kindly inquire about the current status of your review regarding our submission. **We sincerely hope that our responses have adequately addressed your concerns. Furthermore, we are eager to address any additional queries you might have, which will enable us to enhance our work further.**
> > > >
> > > > Once again, thank you for your guidance and support.
> > > >
> > > > Best, Authors

---

### Official Review · Reviewer_zWrq · 2025-10-30

**Soundness:** 3
**Presentation:** 3
**Contribution:** 2
**Rating:** 6
**Confidence:** 4

**Summary:**

This paper proposes LogitSpec to predict the next next token using the last logits, aiming to achieve a speedup in retrieval-based SD by leveraging the potential of the top-k candidates from the last logits.

**Strengths:**

1. This paper presents a clearly written narrative and well-structured exposition that is easy to follow.
2. This paper proposes LogitSpec to predict the next next token using the last logits, aiming to achieve a speedup in retrieval-based SD by leveraging the potential of the top-k candidates from the last logits.

**Weaknesses:**

1. I think the statement “a larger m-gram leads to more precise matches but lower match probabilities” would be clearer with a concrete example to illustrate the difference between being matched and being accurate in retrieval-based SD.

2. I suggest explicitly claiming—e.g., in §4.2—the distinction between this paper’s draft tree and common tree verification. Prior trees are built to compensate for a weak draft model by amplifying diversity at the current decoding step, whereas your tree is constructed with only the target model, treating the current prediction as ground truth to anticipate subsequent steps.

3. In §4.2 I don’t quite understand the claim: “if the rank of the next-next token is below 8, we preserve 4 tokens; if the rank of the next-next token is below 32, we preserve 3 tokens; otherwise we only preserve the speculated next-next token itself.” Does “rank” mean the position of the next-next token within the top-K list (K by default 60)? Also, in retrieval, the m-gram length m is defaulted to 3—so why are there cases where 4 tokens are preserved?

4. A major concern is that reporting results on the relatively outdated Vicuna series as the main table seems suboptimal, even if some newer-model results are provided in the appendix.

**Questions:**

Please refer to the weaknesses.

---

> ### Author Response · Authors · 2025-11-15
> **Response to Reviewer zWrq (part 1 / 2)**
>
> We thank the reviewer for the insightful and valuable comments. We respond to each comment as follows and sincerely hope that our rebuttal could properly address your concerns. If so, we would deeply appreciate it if you could raise your score. If not, please let us know your further concerns, and we will continue actively responding to your comments and improving our submission.
>
> > 1. I think the statement “a larger m-gram leads to more precise matches but lower match probabilities” would be clearer with a concrete example to illustrate the difference between being matched and being accurate in retrieval-based SD.
>
> Thank you for this excellent suggestion. We provide the following concrete example to illustrate the trade-off between match precision and match probability:
>
> **Prompt:** *"Find the area of the triangle. Hint: the area formula is base times height divided by two."*
>
> **Generated so far:** *"To find the"*
>
> **Next token:** *"area"*
>
> Table R1: A concrete example to illustrate the trade-off between match precision and match probability.
>
> | m    | Query (m-gram)               | Matched Reference                     | Retrieved Draft Tokens | Result          |
> | ---- | ---------------------------- | ------------------------------------- | ---------------------- | --------------- |
> | 2    | ["the", "area"]              | "the area **[formula is base]**"      | "formula is base"      | ❌ Wrong match   |
> | 3    | ["find", "the", "area"]      | "Find the area **[of the triangle]**" | "of the triangle"      | ✓ Correct match |
> | 4    | ["To", find", "the", "area"] | None                                  | None                   | ❌ No match      |
>
> This example demonstrates that:
>
> - **m=2 (shorter):** High match probability but retrieves irrelevant tokens ("formula is base" instead of "of the triangle")
> - **m=3 (balanced):** Finds the correct reference with reasonable match probability
> - **m=4 (longer):** Too specific, fails to find any match
>
> This example concretely shows that larger m-grams tend to produce more precise matches when they exist, but they are also less likely to be matched at all.
>
> > 2. I suggest explicitly claiming—e.g., in §4.2—the distinction between this paper’s draft tree and common tree verification. Prior trees are built to compensate for a weak draft model by amplifying diversity at the current decoding step, whereas your tree is constructed with only the target model, treating the current prediction as ground truth to anticipate subsequent steps.
>
> Thank you for pointing out this important conceptual distinction. We agree that it should be stated more explicitly.
>
> Prior tree-based speculative decoding methods typically use a **separate, weaker draft model** to generate multiple diverse candidates at the *current* decoding step, and then verify these branches using the stronger target model. The tree structure is mainly a way to compensate for the draft model’s limited accuracy by amplifying local diversity.
>
> In contrast, **LogitSpec builds the draft tree using only the target model itself**:
>
> - We treat the target model’s current prediction as the ground truth and use its last logits to *speculate* future tokens (e.g., next-next tokens).
> - These speculated tokens, together with retrieval results, are used to grow a multi-step draft tree that is directly verified by the **same** target model via Tree Attention, without any auxiliary draft model.
>
> In the revision, we will update §4.2 to explicitly emphasize this difference, for example by adding a sentence along the lines of:
>
> `“Unlike prior tree-based speculative decoding approaches that rely on a separate draft model to generate diverse local candidates, our draft tree is constructed purely from the target model’s own predictions (via last-logit speculation) and is directly verified by the target model using Tree Attention.”`

---

> > ### Author Response · Authors · 2025-11-15
> > **Response to Reviewer zWrq (part 2 / 2)**
> >
> > > 3. In §4.2 I don’t quite understand the claim: “if the rank of the next-next token is below 8, we preserve 4 tokens; if the rank of the next-next token is below 32, we preserve 3 tokens; otherwise we only preserve the speculated next-next token itself.” Does “rank” mean the position of the next-next token within the top-K list (K by default 60)? Also, in retrieval, the m-gram length m is defaulted to 3—so why are there cases where 4 tokens are preserved?
> >
> > We apologize for the confusion and offer two clarifications:
> >
> > **"Rank"** refers to the position of a speculated next-next token within the **top-K list** (K=60 by default) of the last logit. For instance, rank=5 means the token is the 5th most probable next-next token according to the last logit.
> >
> > **Why 4 tokens can be preserved with m=3:** The m-gram (m=3) is the **query key** for retrieval, while the preserved tokens are the **retrieved n-gram values** (subsequent tokens after the match). These are different:
> >
> > - **m-gram (query):** Last 3 tokens used for matching (e.g., ["find", "the", "area"])
> > - **n-gram (result):** Subsequent tokens after the matched position (e.g., "of the triangle. Hint:")
> >
> > Since the retrieval corpus (prompt + generated tokens) can contain long sequences, a single m-gram query can retrieve many subsequent tokens (potentially 10-20+). The pruning strategy limits this to **4, 3, or 1 token(s)** based on the confidence (rank) of the speculated next-next token. We will clarify this distinction in the revised **Section 4.2** to avoid confusion.
> >
> >
> >
> > > 4. A major concern is that reporting results on the relatively outdated Vicuna series as the main table seems suboptimal, even if some newer-model results are provided in the appendix.
> >
> > We appreciate the concern about using the relatively outdated Vicuna series as the main table. Our primary reason for doing so is **comparability with a wide range of baselines**:
> >
> > - Many existing speculative decoding baselines we compare against (REST and Lookahead) modify the underlying modeling files (`modeling_*.py`) and currently provide stable implementations only for the Vicuna / Llama-2 family.
> > - For newer models such as Llama-3, several of these baselines are not available or would require non-trivial re-engineering of their codebases, which would make the comparison less fair or less reproducible.
> >
> > To balance **baseline coverage** and **modern model evaluation**, we adopted the following strategy:
> >
> > 1. Use Vicuna as the main table where we can include a **comprehensive set of baselines** under a unified and well-tested implementation stack.
> > 2. Provide additional experiments on **Llama-3** in the appendix to demonstrate that LogitSpec also brings consistent gains on more advanced reasoning models.
> >
> > Following the reviewer’s suggestion, in the revised manuscript we will **promote key results on Llama-3 from the appendix into the main text** (e.g., a table summarizing LogitSpec vs. strong baselines on AIME/LongBench/MATH with modern models). In addition, we include comparisons against recent plug-and-play SOTA methods such as **SAM Decoding** and **Token Recycling** on these modern benchmarks in Table R2, where LogitSpec consistently achieves higher MAT and speedup. This will better highlight the practical relevance of LogitSpec on up-to-date LLMs while still preserving fair and reproducible comparisons with a broad set of existing speculative decoding methods.
> >
> > Table R2: Comparison with Token Recycling and SAM Decoding on advanced benchmarks using Llama-3.1-8B-Instruct.
> >
> > |           | AIME MAT | AIME speedup | LongBench MAT | LongBench Speedup | MATH MAT | MATH Speedup | Avg MAT  | Avg Speedup |
> > | --------- | -------- | ------------ | ------------- | ----------------- | -------- | ------------ | -------- | ----------- |
> > | AR        | 1.00     | 1.00x        | 1.00          | 1.00x             | 1.00     | 1.00x        | 1.00     | 1.00x       |
> > | SAM       | 3.11     | 2.73x        | 3.01          | 2.04x             | 2.66     | 2.26x        | 2.92     | 2.34x       |
> > | Recycling | 3.10     | 2.82x        | 2.96          | 1.97x             | 2.80     | 2.40x        | 2.96     | 2.40x       |
> > | LogitSpec | **3.57** | **3.09x**    | **3.08**      | **2.10x**         | **3.05** | **2.56x**    | **3.23** | **2.58x**   |
> >
> >
> > ---
> >
> > **We humbly hope our response has addressed your concerns. If you have any additional concerns or comments that we may have missed in our responses, we would be most grateful for any further feedback from you to help us further enhance our work.**

---

> > > ### Author Response · Authors · 2025-11-26
> > > **We are looking forward to your further comments and/or questions.**
> > >
> > > Dear Reviewer zWrq,
> > >
> > > We deeply appreciate your valuable feedback and the time you've taken to review our work, especially during this busy period.
> > >
> > > We are reaching out to kindly inquire about the current status of your review regarding our submission. **We sincerely hope that our responses have adequately addressed your concerns. Furthermore, we are eager to address any additional queries you might have, which will enable us to enhance our work further.**
> > >
> > > Once again, thank you for your guidance and support.
> > >
> > > Best, Authors

---

### Official Review · Reviewer_Zriw · 2025-10-30

**Soundness:** 2
**Presentation:** 3
**Contribution:** 2
**Rating:** 2
**Confidence:** 4

**Summary:**

This paper introduces LogitSpec, which utilizes the top-k candidates in the logit of the last token as an extra content of drafts to assist the retrieval-based speculative decoding (SD). Concretely, it uses the top-60 tokens in the last token logits as the "next next token" speculation and conducts retrieval for m-gram spans with both the next token and the "next next token". Experiments demonstrate that LogitSpec can achieve up to 2.61x speedup and 3.28 mean accepted tokens per decoding step.

**Strengths:**

1. This work proposes a new perspective to improve retrieval-based speculative decoding by utilizing the top-k tokens in the last token logits as an additional draft source.
2. The authors conduct comprehensive experiments on a wide range of text generation benchmarks. Experiments demonstrate that LogitSpec can achieve up to 2.61x speedup and 3.28 mean accepted tokens per decoding step.
3. The paper also provides an analysis of module contributions inLogitSpec, which clearly demonstrates that the last logit module contributes to the speedup improvements of SD.
4. This manuscript is well-written with clear demonstrations.

**Weaknesses:**

1. **The necessity of next next token**: My major concern lies in the necessity of introducing the "next next token" as an extra retrieval source.
   - Firstly, most retrieval-based SD methods consider a context window of generated tokens as the source spans, generally 3 or 5. In this view, it seems that introducing the "next next token" merely increases the context window length by 1. However, as the authors noted in Lines 794-799, the parallel consideration of top-60 candidates to construct draft sequences introduces significant extra computational overhead. It is doubtful whether LogitSpec achieves a better efficiency trade-off compared to other retrieval-based SD.
   - The authors preserve 3 retrieved tokens if the rank of the "next next token" is below 32 and only preserve the "next next token" itself if its rank exceeds 32. It means that a large part of draft sequences only consider the "next next token" without retrieval. I wonder whether the sacrifice of "retrieval contents" in the draft sequences is worth it. This heuristic pruning strategy should be further analyzed.
2. **Stronger baselines are needed:** LogitSpec should be compared with stronger baselines such as Token Recycling [1] and SAM Decoding [2]. According to the third-party leaderboard of Spec-Bench [3], both methods, which are also plug-and-play retrieval-based methods, achieve significant efficiency improvements over Lookahead [4] and PLD. LogitSpec should be compared with these SOTA approaches, or combined with them, to better illustrate its superiority.
3. The authors claim that "the MAT of last logit decoding exhibits superior robustness to the task" in Line 249. However, in Fig. 2(b), the "last logit" achieves lower MAT compared to PLD in 4 out of 6 tasks. The wording of "robustness" does not reflect the superiority of the "last logit".
4. This manuscript conducts additional results with advanced reasoning models like Qwen-3-8B. I wonder that the detailed implementation is huggingface transformers or vllm.



[1] Turning Trash into Treasure: Accelerating Inference of Large Language Models with Token Recycling. Luo et al. ACL 2025.

[2] SAM Decoding: Speculative Decoding via Suffix Automaton. Hu et al. ACL 2025.

[3] Unlocking Efficiency in Large Language Model Inference: A Comprehensive Survey of Speculative Decoding. Xia et al. Findings of ACL 2024.

[4] Break the Sequential Dependency of LLM Inference Using Lookahead Decoding. Fu et al. ICML 2024.

**Questions:**

Please refer to the Weaknesses part above.

---

> ### Author Response · Authors · 2025-11-15
> **Response to Reviewer Zriw (Part 1 / 3)**
>
> We thank the reviewer for the insightful and valuable comments. We respond to each comment as follows and sincerely hope that our rebuttal could properly address your concerns. If so, we would deeply appreciate it if you could **raise your score ( 2: reject, not good enough).** If not, please let us know your further concerns, and we will continue actively responding to your comments and improving our submission.
>
> > 1. **The necessity of next next token**: My major concern lies in the necessity of introducing the "next next token" as an extra retrieval source.
> >    - Firstly, most retrieval-based SD methods consider a context window of generated tokens as the source spans, generally 3 or 5. In this view, it seems that introducing the "next next token" merely increases the context window length by 1. However, as the authors noted in Lines 794-799, the parallel consideration of top-60 candidates to construct draft sequences introduces significant extra computational overhead. It is doubtful whether LogitSpec achieves a better efficiency trade-off compared to other retrieval-based SD.
>
> We thank the reviewer for raising this critical point. We first address the concerns regarding necessity and overhead of the "next next token" (**NNT**).
>
> 1. **More than “context + 1”: enabling a tree rather than a chain.**
>
>    Retrieval-based SD methods such as PLD can only form drafts based on *already generated* tokens. In many low-repetition scenarios, the currently generated suffix has not appeared before, and pure n-gram matching fails to find useful spans (e.g., “the” appears often, but “the capital” only becomes a distinctive key once “capital” is predicted).
>
>    In LogitSpec, we use the last logits to *speculate* the next-next token and then use this speculative token as part of the retrieval key. This allows us to expand the first layer of the draft **tree** from the model’s own prediction of future tokens, rather than only from the realized history. As a result, the same draft budget (up to 60 tokens) is organized into a wider tree instead of a single chain, and we empirically observe a higher fraction of accepted tokens for the same verification cost.
>
> 2. **Parallel verification of up to 60 draft tokens is almost free on modern GPUs.**
>
>    Recent work (e.g., OPT-Tree [1] ) and our measurements on A100/H100 show that decoding is largely **memory-bound**: verifying 1, 16, 32, or even 60 draft tokens in a single step leads to only modest latency differences. In other words, within this range we can increase the draft capacity without linearly increasing wall-clock cost. This is also why many existing speculative decoding methods (Eagle, Medusa, Token Recycling, etc.) adopt a draft capacity around 60–80 tokens. In our own ablations (Appendix E.5), we find that capacities around 64 indeed maximize speedup.
>
> 3. **Better MAT–efficiency trade-off than chain-based PLD.**
>
>    Compared with PLD, LogitSpec uses the next-next speculation to *densely populate* this draft tree, leading to more verified tokens per step without noticeably increasing verification overhead. Under the same draft budget (maximum of 60 tokens), LogitSpec consistently achieves higher MAT and overall GPU utilization than PLD across our benchmarks (see Table 1 and Appendix E), showing that the additional speculative branching is justified by the efficiency gains.
>
> In summary, LogitSpec's NNT-based tree expansion allows it to fully utilize the parallel verification capacity that PLD's chain-based approach leaves idle, resulting in higher MAT and superior speedup.
>
> [1] Wang, Jikai, Yi Su, Juntao Li, Qingrong Xia, Zi Ye, Xinyu Duan, Zhefeng Wang, and Min Zhang. "Opt-tree: Speculative decoding with adaptive draft tree structure." *Transactions of the Association for Computational Linguistics* 13 (2025): 188-199.

---

> > ### Author Response · Authors · 2025-11-15
> > **Response to Reviewer Zriw (Part 2 / 3)**
> >
> > > - The authors preserve 3 retrieved tokens if the rank of the "next next token" is below 32 and only preserve the "next next token" itself if its rank exceeds 32. It means that a large part of draft sequences only consider the "next next token" without retrieval. I wonder whether the sacrifice of "retrieval contents" in the draft sequences is worth it. This heuristic pruning strategy should be further analyzed.
> >
> > We agree that the pruning strategy represents a trade-off between tree capacity and prediction accuracy. Our design is motivated by **Figure 2(a)**: tokens ranked higher in the last logit have significantly higher acceptance probability. Therefore, our heuristic allocates more retrieval capacity (i.e., longer retrieved n-grams) to these high-potential candidates. To stay within the optimal total capacity (K=64), we must prune the capacity for lower-ranked (and less likely) candidates, which is why those with rank > 32 are not used for retrieval.
> >
> > We provided a detailed ablation on this in **Appendix E.4 (Table 13)** , comparing our heuristic to a different rank-based strategy and an average-allocation strategy:
> >
> > - **Ablation Strategy 1 (rank-based):** Rank <4 → 5 tokens; <8 → 4 tokens; <16 → 3 tokens; <32 → 2 tokens; else 1 token
> > - **Ablation Strategy 2 (uniform):** 4 tokens for all next-next tokens
> > - **Original strategy:** Rank <8 → 4 tokens; <32 → 3 tokens; else 1 token
> >
> > The results show that all strategies yield similar performance, and our current scheme achieves slightly higher MAT and speedup than the uniform allocation (relative differences are modest). We also emphasize that the pruning strategy is not our key contribution.
> >
> > > 2. **Stronger baselines are needed:** LogitSpec should be compared with stronger baselines such as Token Recycling [1] and SAM Decoding [2]. According to the third-party leaderboard of Spec-Bench [3], both methods, which are also plug-and-play retrieval-based methods, achieve significant efficiency improvements over Lookahead [4] and PLD. LogitSpec should be compared with these SOTA approaches, or combined with them, to better illustrate its superiority.
> >
> > We thank the reviewer for this valuable suggestion. We have conducted **new experiments** comparing LogitSpec against the strong baselines mentioned (Token Recycling [1] and SAM Decoding [2]) on several advanced benchmarks (AIME, LongBench, and MATH) using Llama-3.1-8B-Instruct in Table R1.
> >
> > The results, shown in the table below, demonstrate LogitSpec's superior performance. LogitSpec consistently outperforms both methods, achieving **7.5% higher speedup on average** (2.58x vs 2.40x). Notably, on the challenging AIME dataset, LogitSpec achieves 3.09× speedup compared to 2.82× for Token Recycling. We will incorporate these results into the revised paper.
> >
> > Table R1: Comparison with Token Recycling and SAM Decoding on advanced benchmarks using Llama-3.1-8B-Instruct.
> >
> > |           | AIME MAT | AIME speedup | LongBench MAT | LongBench Speedup | MATH MAT | MATH Speedup | Avg MAT  | Avg Speedup |
> > | --------- | -------- | ------------ | ------------- | ----------------- | -------- | ------------ | -------- | ----------- |
> > | AR        | 1.00     | 1.00x        | 1.00          | 1.00x             | 1.00     | 1.00x        | 1.00     | 1.00x       |
> > | SAM       | 3.11     | 2.73x        | 3.01          | 2.04x             | 2.66     | 2.26x        | 2.92     | 2.34x       |
> > | Recycling | 3.10     | 2.82x        | 2.96          | 1.97x             | 2.80     | 2.40x        | 2.96     | 2.40x       |
> > | LogitSpec | **3.57** | **3.09x**    | **3.08**      | **2.10x**         | **3.05** | **2.56x**    | **3.23** | **2.58x**   |

---

> > > ### Author Response · Authors · 2025-11-15
> > > **Response to Reviewer Zriw (Part 3 / 3)**
> > >
> > > > 3. The authors claim that "the MAT of last logit decoding exhibits superior robustness to the task" in Line 249. However, in Fig. 2(b), the "last logit" achieves lower MAT compared to PLD in 4 out of 6 tasks. The wording of "robustness" does not reflect the superiority of the "last logit".
> > >
> > > We appreciate the reviewer pointing out this issue. We acknowledge that PLD achieves higher MAT than last-logit decoding in 4 out of 6 tasks in Figure 2(b). However, our claim of "robustness" refers to **consistent performance across diverse task types**, not absolute superiority.
> > >
> > > The key observation is that PLD's performance is **highly task-dependent**:
> > >
> > > - On Translation (low context repetition): PLD achieves only **1.04× speedup** (MAT ≈ 1.1)
> > > - PLD fails to find matched tokens in **>30% of decoding steps** (Table 4)
> > >
> > > In contrast, last-logit decoding maintains **MAT > 1.5 across all tasks** with minimal variance (1.53-1.68). This consistency demonstrates that the predictive capability of the last logit is an **intrinsic property of LLMs** that generalizes across tasks, unlike retrieval-based methods that depend heavily on context repetition.
> > >
> > > This demonstrates its generalizability to "unlock" the LLM's own predictive power, rather than *only* relying on past n-grams. LogitSpec (our full method) combines this robust baseline with retrieval.
> > >
> > > > 4. This manuscript conducts additional results with advanced reasoning models like Qwen-3-8B. I wonder that the detailed implementation is huggingface transformers or vllm.
> > >
> > > Our implementation is based on Hugging Face `transformers`. To clarify this point (and address a concern from Reviewer jwo1), while our main experiments used `transformers==4.37.1`, the additional results with **Llama-3-8B and Qwen3-8B (Appendix E.1) were run using an upgraded version (4.57.1)**, which is necessary to support these newer models.
> > >
> > > ---
> > >
> > > **We humbly hope our response has addressed your concerns. If you have any additional concerns or comments that we may have missed in our responses, we would be most grateful for any further feedback from you to help us further enhance our work.**

---

> > > > ### Author Response · Authors · 2025-11-26
> > > > **We are looking forward to your further comments and/or questions.**
> > > >
> > > > Dear Reviewer Zriw,
> > > >
> > > > We deeply appreciate your valuable feedback and the time you've taken to review our work, especially during this busy period.
> > > >
> > > > We are reaching out to kindly inquire about the current status of your review regarding our submission. **We sincerely hope that our responses have adequately addressed your concerns. Furthermore, we are eager to address any additional queries you might have, which will enable us to enhance our work further.**
> > > >
> > > > Once again, thank you for your guidance and support.
> > > >
> > > > Best, Authors

---

> > > > > ### Comment · Reviewer_Zriw · 2025-11-27
> > > > >
> > > > > Thank you for the detailed responses. Most of my concerns have been resolved. Considering this, I have increased my rating to 6. My concerns remain on two points: (1) The authors mentioned that the usage of "next next token" expands the first layer of the draft tree from the model’s own prediction. It seems that this augmentation only increases the depth of the tree by 1. I wonder whether the benefit is worth it by introducing this "next next prediction". (2) Qwen3 occupies thousands of reasoning tokens in its thinking process. In such scenarios, vllm could speed up the reasoning process much more than the vanilla transformer + speculative decoding. It would be better to apply logitspec on such advanced inference engines.

---

### Note · Authors · 2026-01-05

I have read and agree with the venue's withdrawal policy on behalf of myself and my co-authors.